# Online Robust Regression via SGD on the $\ell_1$ loss

**Scott Pesme**
EPFL
Lausanne, Switzerland
scott.pesme@epfl.ch

**Nicolas Flammarion**
EPFL
Lausanne, Switzerland
nicolas.flammarion@epfl.ch

## Abstract

We consider the robust linear regression problem in the online setting where we have access to the data in a streaming manner, one data point after the other. More specifically, for a true parameter $\theta^*$, we consider the corrupted Gaussian linear model $y = \langle x, \theta^* \rangle + \varepsilon + b$ where the adversarial noise $b$ can take any value with probability $\eta$ and equals zero otherwise. We consider this adversary to be oblivious (i.e., $b$ independent of the data) since this is the only contamination model under which consistency is possible. Current algorithms rely on having the whole data at hand in order to identify and remove the outliers. In contrast, we show in this work that stochastic gradient descent on the $\ell_1$ loss converges to the true parameter vector at a $\tilde{O}(1/(1-\eta)^2 n)$ rate which is independent of the values of the contaminated measurements. Our proof relies on the elegant smoothing of the non-smooth $\ell_1$ loss by the Gaussian data and a classical non-asymptotic analysis of Polyak-Ruppert averaged SGD. In addition, we provide experimental evidence of the efficiency of this simple and highly scalable algorithm.

## 1 Introduction

Robust learning is a critical field that seeks to develop efficient algorithms that can recover an underlying model despite possibly malicious corruptions in the data. In recent decades, being able to deal with corrupted measurements has become of crucial importance. The applications are considerable, to name a few settings: computer vision [87, 90, 5], economics [85, 73, 92], astronomy [72], biology [89, 76] and above all, safety-critical systems [15, 40, 33].

Linear regression being one of the most fundamental statistical model, the robust regression problem has naturally drawn substantial attention. In this problem, we wish to recover a signal from noisy linear measurements where an unknown proportion $\eta$ has been arbitrarily perturbed. Various models have been proposed to illustrate such contaminations. The broadest is to consider that the adversary is adaptive and is allowed to inspect the samples before changing a fraction $\eta$. In this general framework, exact model recovery is not possible and several robust algorithms have been proposed [19, 49, 22, 68, 77, 17, 55, 54]. The information-theoretic optimal recovery guarantee has recently been reached by [28]. Another model is to consider an oblivious adversary, in this simpler context it is possible to consistently recover the model parameter and several algorithms have been proposed [8, 79].

However, none of these algorithms are suitable for online or large-scale problems [58, 36]. Indeed, all of the suggested algorithms require handling the complete dataset, which is simply unrealistic in such settings. This is a considerable problem when we know that modern problems involve colossal datasets and that current machine learning methods are limited by the computing time rather than the amount of data [12]. Such considerations advocate the necessity of proposing practical, online and highly scalable robust algorithms, hence we ask the following question:

*Can we design an efficient online algorithm for the robust regression problem ?*

In this paper we answer by the affirmative for the online *oblivious response corruption* model where we are given a stream of i.i.d. observation $(x_i, y_i)_{i \in \mathbb{N}}$ from the following generative model:

$$y = \langle x, \theta^* \rangle + \varepsilon + b,$$

where $\theta^* \in \mathbb{R}^d$ is the true parameter we wish to recover, $x$ is the Gaussian feature, $\varepsilon$ is the Gaussian noise of variance $\sigma^2$ and $b$ is an adversarial 'sparse' noise assumed independent of the data $(x, \varepsilon)$ such that $\mathbb{P}(b \neq 0) = \eta$. In order to recover the parameter $\theta^*$, we perform *averaged SGD on the expected $\ell_1$ loss* $\mathbb{E}[|y - \langle x, \theta \rangle|]$. Though this algorithm is very simple, we show that it successively handles the outliers in an online manner and recovers the true parameter at the optimal non-asymptotic convergence rate $\tilde{O}(1/n)$ for any outlier proportion $\eta < 1$. Such an algorithm is useful for abundant practical applications such as : (a) detection of irrelevant measurements and systematic labelling errors [52], (b) real time detection of system attacks such as frauds by click bots [41] or malware recommendation rating-frauds [95], and (c) online regression with heavy-tailed noise [79].

The minimisation problem $\min_{\theta \in \mathbb{R}^d} \mathbb{E}[|y - \langle x, \theta \rangle|]$ is certainly not new and is also known as the Least Absolute Deviations (LAD) problem. While originally suggested by Boscovich in the mid-eighteenth century [10], it first appears in the work of Edgeworth [32]. In contrast with least-squares, there is no closed form solution to the problem and, in addition, the non-differentiability of the $\ell_1$ loss prevents the use of fast optimisation solvers for large-scale applications [88]. However, if successively dealt with, the LAD problem has many advantages. Indeed, the $\ell_1$ loss is well known for its robustness properties [48] and, unlike the Huber loss [46], it is parameter free which makes it more convenient in practice. In this context, using the SGD algorithm in order to solve the LAD problem is a very natural approach. We show in our analysis that, though the $\ell_1$ loss is not strongly convex, averaged SGD recovers a remarkable $O(1/n)$ convergence rate instead of the classical $O(1/\sqrt{n})$ which is ordinary in the non-strongly-convex framework

With a convergence rate depending on the variance as $O(\sigma^2 d/(1 - \tilde{\eta})^2 n)$, the proposed algorithm has several major benefits: a) it is highly scalable and statistically optimal, b) it depends on the outlier contamination through an *effective* outlier proportion $\tilde{\eta}$ strictly smaller than $\eta$, which makes it *adaptive* to the difficulty of the adversary, c) it is relatively insensitive to the ill conditioning of the features and d) it is almost parameter free since it only requires in practice an upper-bound on the covariates' norm. Though the algorithm is simple, its analysis is not and requires several technical manipulations based on recent advances in stochastic approximation [44]. Indeed, in the classical non-strongly-convex framework which we are in, the usual convergence rate is $O(1/\sqrt{n})$ and not $O(1/n)$ as we obtain. Overall our analysis relies on the smoothing of the $\ell_1$ loss by the Gaussian data. This smoothing enables the retrieval of a fast $O(1/n)$ rate thanks to the local strong convexity around $\theta^*$ and to Polyak-Ruppert averaging.

Our paper is organised as follows. We define the problem which we consider in Section 2. We then describe the particular structure our function $f(\theta) := \mathbb{E}[|y - \langle x, \theta \rangle|]$ enjoys in Section 3. Our main convergence guarantee result is given Section 4 followed by the sketch of proof in Section 5. Finally, in Section 6, we provide experimental validation of the performances of our method.

## 1.1 Related work

**Robust statistics.** Classical robust statistics have a long history which begins with the seminal work by Tukey and Huber [82, 46]. They mostly focus on the influence function [43], the asymptotic efficiency [91] as well as on the concept of breakdown point [42, 29] which is the maximal proportion of outliers an estimate can tolerate before it breaks down. However these approaches are purely statistical and the proposed estimators are unfortunately either not computable in polynomial time [70, 71] or purely heuristical [34].

**Recent advances in robust statistics.** [23, 51] are the first to propose robust estimators of the mean that can be computed in polynomial time and that have near optimal sample complexities. This leads to a recent line of work in the computer science community that provides recovery guarantees for a range of different statistical problems such as mean estimation or covariance estimation [24, 27, 25, 50, 45, 77, 17]. The robust linear regression problem is explored under general corruption models in the works of [19, 49, 22, 28, 68, 77, 17, 55, 54]. The broadest and therefore hardest contamination model considers that the adversary has access to all the samples and can arbitrarily contaminate any fraction $\eta$. In this contamination framework the minimax optimal

estimation rate on $\|\hat{\theta} - \theta^*\|$ is $\tilde{O}(\sigma(\eta + \sqrt{d/n}))$ where $\sigma^2$ is the variance of the Gaussian dense noise [18]. In [28] this minimax bound is achieved under the assumption that the covariates follow a centered Gaussian distribution of covariance identity. For a general covariance matrix $H$, the sample complexity becomes $O(d^2/\eta^2)$, however they provide a statistical query lower bound showing that it may be computationally hard to approximate $\theta^*$ with less than $d^2$ samples. We highlight the fact that if the computational issues are put aside, [39] provides for the weaker Huber $\varepsilon$ contamination model[1] a statistically optimal estimator which can however only be computed in exponential time. For more details on the current advances, see the recent survey [26].

**Response corrupted robust regression.** A simpler contamination model is to consider that the adversary can only corrupt the responses and not the features. In this framework two main approaches have been considered.
**a)** The first approach is based on viewing the regression problem as $\min_{\theta, b: \|b\|_0 \leq \eta n} \|y - X\theta - b\|_2$. However this is a non-convex and NP hard problem [78]. In order to deal with the problem, convex relaxations based on the $\ell_1$ loss [86, 64, 63] and second-order-conic-programming (SOCP) [14, 20] have been proposed and studied. Simultaneously, hard thresholding techniques were considered [9, 8, 79]. However all of these approaches rely on manipulating the whole corruption vector $b$ and are therefore not easy to adapt to the online setting.
**b)** The second approach relies on using a so-called robust loss. This is designated as the M-estimation framework [46]: the least-squares problem is replaced by $\min_\theta \mathbb{E}_z[\rho(\theta, z)]$ where the loss function $\rho$ is chosen for its robustness properties. The $\ell_1$ loss or the Huber loss [46] are classical examples of convex robust losses. The Huber loss is essentially an appropriate mix of the $\ell_2$ and the $\ell_1$ losses. On the other hand the Tukey biweight [82] is an example of a non-convex robust loss. The idea behind such losses is to give less weight to the outliers which have large residuals. Asymptotic normality of these M-estimators have been well studied in the statistical literature [47, 6, 57, 65, 84, 83, 80] and their non-asymptotic performance have been recently investigated [56, 94, 59, 48].

We point out that the two mentioned approaches are related since they are duals. Minimising the Huber Loss is equivalent to the $\ell_1$ constrained problem $\min_{\theta, b} \|y - X\theta - b\|_2^2 + \lambda\|b\|_1$ for an appropriate $\lambda$ [37, 30]. The estimation rates of $\|\hat{\theta} - \theta^*\|_2$ in both cases are similar and typically $\tilde{O}(\sigma(\sqrt{\eta} + \sqrt{d/n}))$. These rates were later improved to the optimal rate $\tilde{O}(\sigma(\eta + \sqrt{d/n}))$ in [21]. When the adversary is in addition oblivious, consistency is possible and [80, 8, 79] show that there exists a consistent estimator with error $O(\sigma(1 - \eta)^{-1}\sqrt{d/n})$.

Finally we mention that a different approach to deal with response corruptions is proposed in [13]. They consider the noise as coming from a Student's $t$-distribution and propose an EM-algorithm to learn the number of degrees of freedom. However they do not give any convergence guarantee and each step of their algorithm is computationally heavy, making it inappropriate for large-scale problems.

**Stochastic optimisation.** Stochastic optimisation has been studied in a variety of different frameworks such as that of machine learning [11, 93], optimisation [60] and stochastic approximation [7]. The optimal convergence rates are known since [61]: it is of $O(1/\sqrt{n})$ in the general convex case and is improved to $O(1/\mu n)$ if $\mu$-strong convexity is additionally assumed. These rates are obtained using the SGD algorithm. However the optimal step-size sequences depend on $f$'s convexity properties. The idea of averaging the SGD iterates first appears in the works of [66] and [74]. This method is now referred to as Polyak-Ruppert averaging. Shortly after, [67] provides asymptotic normality results on the probability distribution of the averaged iterates. This result is later generalised in [3] where non-asymptotic guarantees are provided. In the smooth framework, the advantages of averaging are numerous. Indeed, it improves the global convergence rate and leads to the statistically optimal asymptotic variance rate which is independent of the conditioning $\mu$ [4]. Moreover it has the significant advantage of providing an algorithm that adapts to the difficulty of the problem: with averaging, the same step-size sequence leads to the optimal convergence rate whether the function is strongly convex or not. Averaging also displays another important property which will prove to be particularly relevant in our work: it leads to fast convergence rates in some cases where the function is only locally strongly convex around the solution, as for logistic regression [2].

## 2 Problem formulation

Consider we have a stream of independent and identically distributed data points $(x_i, y_i)_{i \in \mathbb{N}}$ sampled from the following linear model:

(**A.1**)
$$y_i = \langle x_i, \ \theta^* \rangle + \varepsilon_i + b_i,$$

where $\theta^* \in \mathbb{R}^d$ is a parameter we wish to recover. The noises $\varepsilon_i$ are considered as 'nice' noise of finite variance $\sigma^2$. In contrast the outliers $b_i$ can be any adversarial 'sparse' noise. In the online setting we define a sparse random variable as a random variable that equals $0$ with probability $(1 - \eta) \in (0, 1]$.

We investigate in this work the $\ell_1$ minimisation problem (a.k.a least absolute deviation):

$$\min_{\theta \in \mathbb{R}^d} f(\theta) := \mathbb{E}\left[ |y - \langle x, \ \theta \rangle| \right], \tag{1}$$

using the stochastic gradient descent algorithm [69] defined by the following recursion initialised at $\theta_0 \in \mathbb{R}^d$:

$$\theta_n = \theta_{n-1} + \gamma_n \mathrm{sgn}\left( y_n - \langle x_n, \ \theta_{n-1} \rangle \right) x_n, \tag{2}$$

where $(\gamma_n)_{n \geq 1}$ is a positive sequence named step-size sequence. In this paper we mostly consider the averaged iterate $\bar{\theta}_n = \frac{1}{n} \sum_{i=0}^{n-1} \theta_i$ which can easily be computed online as $\bar{\theta}_n = \frac{1}{n} \theta_{n-1} + \frac{n-1}{n} \bar{\theta}_{n-1}$. Note that SGD is an extremely simple and highly scalable streaming algorithm. There are no parameters to tune, we will see further that a step-size sequence of the type $\gamma_n = 1/R^2 \sqrt{n}$ leads to a good convergence rate.

We make here the following assumptions:

(**A.2**) **Gaussian features.** The features $x$ are centered Gaussian $\sim \mathcal{N}(0, H)$ where $H$ is a positive definite matrix. We denote by $\mu$ its smallest eigenvalue and $R^2 = \mathrm{trace}(H)$.

(**A.3**) **Independent Gaussian dense noise.** The dense noise $\varepsilon$ is a centered Gaussian $\mathcal{N}(0, \sigma^2)$ where $\sigma > 0$ and $\varepsilon$ is independent of $x$.

(**A.4**) **Independent sparse adversarial noise.** The adversarial noise $b$ is independent of $(x, \varepsilon)$, it satisfies $\mathbb{E}[|b|] < +\infty$ and $\mathbb{P}(b \neq 0) = \eta \in [0, 1)$. We denote by $\tilde{\eta} = \eta \cdot (1 - \mathbb{E}_b[\exp(-\frac{b^2}{2\sigma^2}) \mid b \neq 0]) \in [0, \eta)$.

Under these assumptions $f'(\theta^*) = -\mathbb{E}\left[ \mathrm{sgn}\left( \varepsilon + b \right) x \right] = -\mathbb{E}\left[ \mathrm{sgn}\left( \varepsilon + b \right) \right] \mathbb{E}\left[ x \right] = 0$. It therefore makes sense to minimise $f$ in order to recover the parameter $\theta^*$. Note that contrary to least-squares where a finite mean and variance are required, we do not make any assumptions on the moments of the noise $b$. The Gaussian assumptions on data are technical and made for simplicity. However the continuous aspect of the features and the noise is essential in order to have a differentiable loss $f$ after taking the expectation. We point that the Gaussian assumptions is a classical assumption which is also made in the works of [9, 8, 79, 48, 28]. However we do not make any additional hypothesis on matrix $H$ concerning its conditioning. We point out that in our framework we must have $\varepsilon$ and the adversarial noise $b$ independent of $x$ in order to have $\mathrm{argmin}_{\theta \in \mathbb{R}^d} \mathbb{E}\left[ |y - \langle x, \ \theta \rangle| \right] = \theta^*$. The parameter $\tilde{\eta} \in [0, \eta)$ we introduce in Assumption A.4 is the *effective outlier proportion*. This quantity expresses the pertinent corruption proportion and will prove to be more relevant than $\eta$. Indeed, notice that in the simple case where $\mathbb{P}(b \ll \sigma) \sim 1$ then $\mathbb{E}_b[\exp(-b^2/\sigma^2)] \sim 1$ and $\tilde{\eta} \sim 0$. Intuitively in this situation it makes sense to have an effective outlier corruption close to zero: if $b \ll \sigma$ a.s., then the $b_i$'s do not disturb the recursion and can therefore be considered as non-adversarial noises. This last observation is however only valid in the oblivious framework.

## 3 Gaussian smoothing and structure of the objective function $f$

In this section we show that $f$ and its derivatives enjoy nice properties and have closed forms which prove to be useful in analysing the SGD recursion Section 5. We fist show that though the $\ell_1$ loss is not smooth, it turns out that by averaging over the continuous Gaussian features $x$ and noise $\varepsilon$, the expected loss $f$ is continuously differentiable.

**Lemma 1.** *Suppose that (A.1, A.2, A.3, A.4) hold and let* $\mathrm{erf}(\cdot)$ *denote the Gauss error function. Then for all* $\theta \in \mathbb{R}^d$:

$$f(\theta) = \mathbb{E}_b\left[ \sqrt{\frac{2}{\pi}} \sqrt{\sigma^2 + \|\theta - \theta^*\|_H^2} \exp\left( -\frac{b^2}{2(\sigma^2 + \|\theta - \theta^*\|_H^2)} \right) + b\, \mathrm{erf}\left( \frac{b}{\sqrt{2(\sigma^2 + \|\theta - \theta^*\|_H^2)}} \right) \right].$$

We point out that the expectation $\mathbb{E}_b[\cdot]$ is taken only over the outlier distribution, the expectation over the Gaussian features and Gaussian noise having already been taken. The proof relies on noticing that since $b$ is independent of $x$ and $\varepsilon$, given outlier $b$, $y - \langle x, \theta \rangle = \varepsilon + b - \langle x, \theta - \theta^* \rangle$ follows a normal distribution $\mathcal{N}(b, \sigma^2 + \|\theta - \theta^*\|_H^2)$. The absolute value of this Gaussian random variable follows a folded normal distribution whose expectation has a closed form [53]. Lemma 1 exhibits the fact that though the $\ell_1$ loss is not differentiable at zero, taking its expectation over a continuous density makes the expected loss $f$ continuously differentiable. This is reminiscent of Gaussian smoothing used in gradient-free and non-smooth optimisation [62, 31].

In the absence of contamination, i.e., when $b = 0$ almost surely, the function simplifies to the pseudo-Huber loss function [16] with parameter $\sigma$, $f(\theta) = \sqrt{2/\pi}(\sigma^2 + \|\theta - \theta^*\|_H^2)^{1/2}$. More broadly, notice that $f(\theta) \sim \sqrt{2/\pi}\|\theta - \theta^*\|_H$ when $\|\theta - \theta^*\|_H \to +\infty$ and $f(\theta) - f(\theta^*) \sim \frac{1-\tilde{\eta}}{\sqrt{2\pi}\sigma}\|\theta - \theta^*\|_H^2$ for $\|\theta - \theta^*\|_H \ll \sigma$. This highlights the *quadratic* behaviour of $f$ around the solution $\theta^*$ and its *linear* behaviour far from it. This shows that though $f$ is not strongly convex on $\mathbb{R}^d$, it is locally strongly convex around $\theta^*$. Actually the two criteria $f(\theta) - f(\theta^*)$ and $\|\theta - \theta^*\|_H$ are closely related and we show that the prediction error $\|\theta - \theta^*\|_H^2$ is in fact $O\left(f(\theta) - f(\theta^*) + (f(\theta) - f(\theta^*))^2\right)$ (see Lemma 12 in Appendix).

The next lemma exhibits the fact that $f'$ has a surprisingly neat structure.

**Lemma 2.** *Suppose that (A.1, A.2, A.3, A.4) hold. Then for every $\theta \in \mathbb{R}^d$:*

$$f'(\theta) = \alpha(\|\theta - \theta^*\|_H)\, H(\theta - \theta^*),$$

*with $\alpha(z) = \sqrt{\frac{2}{\pi}} \frac{1}{\sqrt{\sigma^2 + z^2}} \mathbb{E}_b\left[\exp\left(-\frac{b^2}{2(\sigma^2 + z^2)}\right)\right]$ for $z \in \mathbb{R}$.*

This result is immediately obtained by deriving $f$'s closed form. The gradient $f'$ benefits from a very specific structure: it is exactly the gradient of the $\ell_2$ loss with a scalar factor in front which depends on $\sigma$, the outlier distribution and the prediction loss $\|\theta - \theta^*\|_H$. Note that the gradient is proportional to $H$, this proves to be useful in our analysis. Also note that the gradients are uniformly bounded over $\mathbb{R}^d$ since $\|f'(\theta)\|_2^2 \leq R^2$. This fact, already predictable from the expression of the stochastic gradients, stands in sharp contrast with the $\ell_2$ loss and illustrates the $\ell_1$ loss's robustness property.

Finally the following lemma highlights $f$'s local strong convexity around $\theta^*$ which is essential to obtain the $O(1/n)$ convergence rate.

**Lemma 3.** *Suppose that (A.1, A.2, A.3, A.4) hold. Then*

$$f''(\theta^*) = \sqrt{\frac{2}{\pi}} \frac{1 - \tilde{\eta}}{\sigma}\, H.$$

Lemma 3 shows that $f$ is locally strongly convex around $\theta^*$ with local strong convexity constant $\sqrt{2/\pi}\frac{(1-\tilde{\eta})\mu}{\sigma}$. Note that without additional gaussian noise $\varepsilon$, which corresponds to $\sigma \to 0$, then there is no smoothing of the $\ell_1$ loss anymore and the problem becomes non-smooth.

## 4 Convergence guarantee

The nice properties the function $f$ enjoys enables a clean analysis of the SGD recursion with decreasing step sizes. The convergence rate we obtain on the averaged iterate $\bar{\theta}_n = \frac{1}{n}\sum_{i=0}^{n-1}\theta_i$ is given in the following theorem.

**Theorem 4.** *Let (A.1, A.2, A.3, A.4) hold and consider the SGD iterates following Eq. (2). Assume $\gamma_n = \frac{\gamma_0}{\sqrt{n}}$. Then for all $n \geq 1$:*

$$\mathbb{E}\left[\|\bar{\theta}_n - \theta^*\|_H^2\right] = O\left(\frac{\sigma^2 d}{(1-\tilde{\eta})^2 n}\right) + \tilde{O}\left(\frac{\|\theta_0 - \theta^*\|^4}{\gamma_0^2(1-\tilde{\eta})^2 n}\right) + \tilde{O}\left(\frac{\gamma_0^2 R^4}{(1-\tilde{\eta})^2 n}\right)$$

$$+ \tilde{O}\left(\frac{\sigma^2}{\gamma_0^2 \mu^2 (1-\tilde{\eta})^3 n^{3/2}}\left(\frac{\|\theta_0 - \theta^*\|^2}{\gamma_0} + \gamma_0 R^2\right)\right).$$

For clarity the exact constants are not given here but can be found in the Appendix. Note that the result is given in terms of the Mahalanobis distance associated with the covariance matrix $H$ which

corresponds to the classical prediction error. The overall bound has a dependency in the number of iterations of $O(1/n)$, this is optimal for stochastic approximation even with strong-convexity [61]. We also point out that by a purely naive analysis that does not exploit the specificities of our problem we could easily obtain a $O(1/\sqrt{n})$ rate, which is the common rate for non-strongly-convex functions.

Notice that the convergence rate is not influenced by the magnitude of the outliers but only by their *effective* proportion $\tilde{\eta} \in [0, \eta)$ which is, as we could anticipate, more relevant than $\eta$. This effective outlier proportion can be considerably smaller than $\eta$ if a portion of the outliers behave correctly. Therefore the algorithm *adapts* to the difficulty of the adversary. We also point out that we recover the same dependency in the proportion of outliers as in [80, 8, 79] but with $\tilde{\eta}$ in our case, which is strictly better. The question of the optimality of the $1/(1-\tilde{\eta})^2$ constant is unknown and is an interesting open problem, a trivial lower bound being $1/(1-\tilde{\eta})$. Furthermore, in the finite horizon framework where we have $N$ samples, the breakdown point we obtain is $\tilde{\eta} = 1 - \Omega(\frac{\ln^{3/2} N}{\sqrt{N}})$. This is better than what is obtain in [79] ( $1 - \Omega(\frac{1}{\ln N})$ ) and the same (to a $\ln$ factor) as the asymptotic result from [80].

We now take a closer look at each term. The first term is the dominant variance term and is of the form $\frac{\sigma^2 d}{n}$. It is statistically optimal in terms of $d$ and $n$ since it is also optimal in the simpler framework where there are no outliers [81]. The second term is the dominant bias term which only depends on the distance between the initial point $\theta_0$ and the solution $\theta^*$. Notice that it is proportional to $\|\theta_0 - \theta^*\|^4$ as in [2], however we believe the dependency in $\|\theta_0 - \theta^*\|$ could be improved to $O(\|\theta_0 - \theta^*\|^2)$ by further exploiting the local strong-convexity property. The third term is a by-product of our analysis and comes from the bound on the norm of the stochastic gradients. We conjecture that it could be possible to get rid of it. We highlight that the three dominant $O(1/n)$ terms are independent of the data conditioning constant $\mu$. Finally, the two last terms are higher order terms that depend on $\mu$, they are correcting terms as in [3] due to the fact that our function is not quadratic. Note that all three dominant $O(1/n)$ terms are independent of $\mu$, this is obtained thanks to $f$'s structure and is due to the fact that $f'$ is proportional to $H$ around $\theta^*$, as in the least-squares framework. We clearly see the benefits of this result in the experiments Section 6: unlike the algorithms from [8, 79] which solve successive least-squares problems and are therefore very sensitive to $H$'s conditioning, our algorithm is way less impacted by the ill conditioning.

We underline the fact that the algorithm is parameter free: neither the knowledge of $\sigma$ nor of the outlier proportion $\eta$ are required. Also, note that there is no restriction on the value of $\gamma_0$ but in practice setting it to $O(1/R^2)$ leads to the best results. We believe that instead of considering the $\ell_1$ loss we could have considered the Huber loss and followed the same technical analysis to obtain a $O(1/n)$ rate. However as shown in the experiments Section 6, considering the Huber loss *does not* improve the rate and requires an extra parameter that must be tuned.

## 5   Sketch of proof

We provide an overview of the arguments that constitute the proof of Theorem 4, the full details can be found in the Appendix. We bring out three key steps. First, using the structure of our problem we relate the behaviour of the averaged iterate $\bar{\theta}_n$ to the average of the gradient $\bar{f'}(\theta_n) = \frac{1}{n} \sum_{i=0}^{n-1} f'(\theta_i)$ (see Lemma 14). Then we show that $\bar{f'}(\theta_n)$ converges to 0 at the rate $O(1/n)$ (see Lemma 6). Finally we control the additional terms using generic results that hold for non-strongly-convex and non-smooth functions (see Lemma 7). Technical difficulties arise from (a) the fact that we consider a decreasing step-size sequence, which is necessary in order to have a fully online algorithm and (b) the fact that we want to obtain a leading order term $O(1/n)$ independent of the conditioning constant $\mu$.

First we use $f$'s specific structure to bound the distance between $\bar{\theta}_n$ and the solution $\theta^*$.

**Lemma 5.** *Let (A.1, A.2, A.3, A.4) hold. Then for any sequences $(\theta_i)_{i=0}^{n-1} \in \mathbb{R}^{dn}$ their average $\bar{\theta}_n = \frac{1}{n} \sum_{i=0}^{n-1} \theta_i$ satisfies:*

$$\mathbb{E}\left[\left\|\bar{\theta}_n - \theta^*\right\|_H^2\right] \leq \frac{2\sigma^2}{(1-\tilde{\eta})^2} \mathbb{E}\left[\left\|\frac{1}{n}\sum_{i=0}^{n-1} f'(\theta_i)\right\|_{H^{-1}}^2\right] + \frac{800}{(1-\tilde{\eta})^2}\left(\ln\frac{2}{1-\eta}\right)^2 \mathbb{E}\left[\left(\frac{1}{n}\sum_{k=0}^{n-1}\langle f'(\theta_i),\,\theta_i - \theta^*\rangle\right)^2\right].$$

This result follows from the inequality $\|f'(\theta) - f''(\theta^*)(\theta - \theta^*)\|_{H^{-1}} \leq \frac{20}{\sigma_1}(\ln\frac{2}{1-\eta})\langle f'(\theta),\,\theta - \theta^*\rangle$ (see proof in Appendix) which upper-bounds the remainder of the first-order Taylor expansion of the gradient by $f$'s linear approximation $\langle f'(\theta),\,\theta - \theta^*\rangle$. This inequality is crucial in our analysis

since in the non-strongly-convex framework, the averaged linear approximations always converge to zero while the iterates $\theta_i$ *a priori* do not converge to $\theta^*$. This inequality is highly inspired by the self-concordance property of the logistic loss used in [2] but is simpler in our setting thanks to an ad hoc analysis. Note that the result from Lemma 14 is valid for any sequence $(\theta_i)_{i \geq 0}$ and not only the one issued from the SGD recursion. The two quantities that we therefore need to control are clear. The first one is central as it leads to the final dominant variance term, the second one is technical and less important, it is left for the end of the section. We first show that $\left\| \bar{f}'(\theta_n) \right\|^2$, the square norm of the average of the gradients, converges at rate $O(1/n)$.

**Lemma 6.** *Let (A.1, A.2, A.3, A.4) and consider the SGD iterates following Eq. (2). Assume* $\gamma_n = \frac{\gamma_0}{\sqrt{n}}$. *Then for all* $n \geq 1$ :

$$\mathbb{E}\left[ \left\| \frac{1}{n} \sum_{i=0}^{n-1} f'(\theta_i) \right\|_{H^{-1}}^2 \right] \leq \frac{16d}{n} + \frac{4}{n\gamma_0^2} \mathbb{E}\left[ \|\theta_n - \theta^*\|_{H^{-1}}^2 \right] + \frac{4}{n^2\gamma_0^2} \|\theta_0 - \theta^*\|_{H^{-1}}^2$$

$$+ \frac{4}{n^2} \Big( \sum_{i=1}^{n-1} \mathbb{E}\left[ \|\theta_i - \theta^*\|_{H^{-1}}^2 \right]^{1/2} \Big( \frac{1}{\gamma_{i+1}} - \frac{1}{\gamma_i} \Big) \Big)^2 .$$

The proof technique of the inequality is similar to those of [67, 3, 35] and relies on the classical expansion $\sum_{i=1}^{n} f_i'(\theta_{i-1}) = \sum_{i=1}^{n} \frac{\theta_{i-1} - \theta_i}{\gamma_i}$. We stress out the fact that from here, one could simply choose to upper bound $\mathbb{E}[\|\theta_i - \theta^*\|_{H^{-1}}^2]$ using classical non-strongly convex bounds which would lead to $\mathbb{E}[\|\theta_i - \theta^*\|_{H^{-1}}^2] \leq \|\theta_0 - \theta^*\|_{H^{-1}} + \gamma_0^2 d \ln ei$ (see Appendix for more details). However re-injecting such general bounds into Lemma 6 would lead to a final bound on $\|\bar{\theta}_n - \theta^*\|_H^2$ with a leading bias term $O(1/\mu n)$. In order to get rid of this dependency in $\mu$ we need to exploit $f$'s structure and obtain a tighter bound on $\mathbb{E}[\|\theta_n - \theta^*\|_{H^{-1}}^2]$. In the following lemma we provide a sharper bound on $\mathbb{E}[\|\theta_n - \theta^*\|_H^2]$ as well as give a bound on the second residual term from Lemma 14.

**Lemma 7.** *Let (A.1, A.2, A.3, A.4) and consider the SGD iterates following Eq. (2). Assume* $\gamma_n = \frac{\gamma_0}{\sqrt{n}}$. *Then for all* $n \geq 1$:

$$\mathbb{E}\left[ \Big( \frac{1}{n} \sum_{k=0}^{n-1} \langle f'(\theta_i), \, \theta_i - \theta^* \rangle \Big)^2 \right] \leq \frac{\ln(en)}{n} \left[ \frac{\|\theta_0 - \theta^*\|^2}{\gamma_0} + 6\gamma_0 R^2 \ln(en) \right]^2 ,$$

*and*

$$\mathbb{E}\left[ \|\theta_n - \theta^*\|_H^2 \right] = \frac{3\sigma \ln(en)}{(1-\bar{\eta})\sqrt{n}} \left[ \frac{3\|\theta_0 - \theta^*\|^2}{\gamma_0} + 4\gamma_0 R^2 \ln(en) \right] + O\left( \frac{1}{n} \right).$$

The proof of the first inequality follows [2]. It uses classical moment bounds in the non-strongly-convex and non-smooth case. The proof of the second inequality is more technical. It relies on the fact that $\|\theta_n - \theta^*\|_{H^{-1}}^2$ can be upper bounded by $O([f(\theta_n) - f(\theta^*)] + [f(\theta_n) - f(\theta^*)]^2)$ (see Lemma 12 in the Appendix) which is due to $f$'s particular structure. We then upper bound $[f(\theta_n) - f(\theta^*)]$'s first and second moment. To do so we follow the recent proof techniques on the convergence of the final iterate from [75, 44]. In our framework there are a few additional technical difficulties coming from the fact that: (a) a decreasing step-size sequence is considered, (b) our iterates are not restricted to a predefined bounded set since no projection is used and (c) our gradients are not almost surely bounded but have bounded second moments. We point out that $f$'s local strong convexity around $\theta^*$ is not exploited to prove Lemma 7, hence we could expect a better dependency in $n$ if this local property was appropriately used, we leave this as future work.

Combining Lemma 7 with Lemma 6 and injecting into Lemma 14 concludes the proof.

## 6 Experiments

In this section we illustrate our theoretical results. We consider the experimental framework of [79] using synthetic datasets. The inputs $x_i$ are i.i.d. from $\mathcal{N}(0, H)$ where $H$ is either the identity matrix (conditioning $\kappa = 1$) or a p.s.d matrix with eigenvalues $(1/k)_{1 \leq k \leq d}$ and random eigenvectors ($\kappa = 1/d$). The outputs $y_i$ are generated following $y_i = \langle x_i, \, \theta^* \rangle + \varepsilon_i + b_i$ where $(\varepsilon_i)_{1 \leq i \leq n}$ are i.i.d. from $\mathcal{N}(0, \sigma^2)$ and the $b_i$'s are defined according to the following contamination model: for

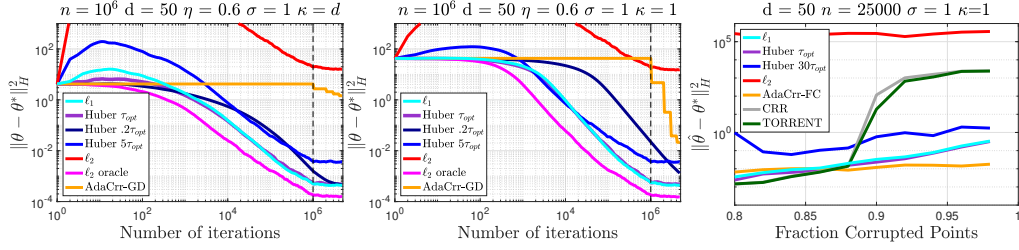

Figure 1: Online robust regression on synthetic data. Left and middle: Convergence rates for a fixed $\eta$ and for two different conditioning of $H$. The dashed line marks the first pass over the data. Right: Estimation performance when varying the portion of corruption $\eta$.

$\eta > 0.5$, a set of $n/4$ corruptions are set to 1000, another $n/4$ are set to $\sqrt{1000}$ and the rest (to reach proportion $\eta > 0.5$) are sampled from $\mathcal{U}([1, 10])$. All results are averaged over five replications.

**Online robust regression.** We plot the convergence rate of averaged SGD on different loss functions: the $\ell_1$ loss, the $\ell_2$ loss and the Huber loss for which we consider various parameters. We also consider the AdaCRR-GD algorithm from [79]. These curves are compared to an oracle algorithm which corresponds to least-squares regression using constant step-size averaged SGD [4] and where all the corrupted points have been discarded (hence a rate of $O(1/(1 - \eta)n)$ ). Since AdaCRR-GD is an offline algorithm that needs all the data to perform a single gradient step we let all algorithms perform 5 passes over the dataset (passes without replacement). In the SGD setting this corresponds to a total of $5n$ iterations. On the plots we represent by a vertical dashed line the first effective pass over the dataset. Figure 1 in the left and middle plots are shown the experimental results for two different conditioning of $H$: $\kappa = 1$ and $\kappa = 1/d$. Notice that independently of the conditioning our algorithm converges at rate $O(1/n)$ and almost matches the performance obtained by the oracle algorithm. Using the Huber loss leads to mixed results: if the parameter is well tuned to $\tau_{opt}$ then the performance is similar to that of the $\ell_1$ loss, but if the parameter is set too large ($5\tau_{opt}$) then the convergence is slow and ends at a sup-optimal point, if it set too small ($.2\tau_{opt}$) the convergence is slow. Indeed the Huber loss with parameter $\tau$ is equivalent to $\tau\|\cdot\|_1$ when the parameter $\tau$ goes to 0. Therefore doing SGD on the Huber loss for $\tau \to 0$ is equivalent to performing SGD on the $\ell_1$ loss with the smaller step-size sequence $(\tau\gamma_n)_{n\geq0}$. On the other hand, SGD on the $\ell_2$ loss is as predicted not competitive at all. AdaCRR-GD needs to wait a full pass before performing one single step and is in all cases much slower than SGD. Moreover notice that AdaCRR-GD is very sensitive to the conditioning of the covariance matrix: the convergence is much slower for a badly conditioned problem. Indeed in this case the convergence of the gradient descent subroutine used in the algorithm becomes sublinear and it significantly degrades the overall performance. On the other hand the performance of SGD on the $\ell_1$ loss is not affected by the conditioning.

**Breakdown point and recovery guarantees.** In this setting, the number of samples $n$ is fixed and we modify the outlier proportion $\eta$. We compare our algorithm to different baselines: $\ell_2$ regression, Huber regression with a well tuned parameter $\tau_{opt}$ and with a larger parameter $30\tau_{opt}$, Torrent [9], CRR [9], and AdaCRR [79]. The details on their implementation are provided in the Appendix. The results are shown Figure 1, right plot. Notice that averaged SGD on the $\ell_1$ obtains comparable results to Huber regression with parameter $\tau_{opt}$ and to AdaCRR, this without having any hyperparameter to tune. Note also that if the parameter of the Huber loss is set too high then the performance is degraded. The other methods are as expected not competitive.

## 7 Conclusion

In this paper, we studied the response robust regression problem with an oblivious adversary. We showed that by simply performing SGD with Polyak-Ruppert averaging on the $\ell_1$ loss $\mathbb{E}[|y - \langle x, \theta \rangle]$ we successively recover the parameter $\theta^*$ with an optimal $O(1/n)$ rate. The experimental results on synthetic data shows the superiority of our algorithm and its clear advantage for high-scale and online settings.

There are several interesting future directions to our work. One would be to consider other corruption models in the online setting. It would also be interesting to see if we can combine our approach with [1, 38] in order to get results in the case where $\theta^*$ is sparse.

# 8 Broader Impact

As discussed in the introduction, the algorithm we propose can be useful in many practical applications such as : (a) detection of irrelevant measurements and systematic labelling errors [52], (b) detection of system attacks such as frauds by click bots [41] or malware recommendation rating-frauds [95], and (c) online regression with heavy-tailed noise [79].

## Footnotes

[1]This contamination model is weaker because the adversary is oblivious of the uncorrupted samples

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
