[Supplementary Material]

## A  Higher-order moment bounds

In this section we prove classical moment bounds on the SGD iterates following eq. (2) with the decreasing step-size sequence $\gamma_n = \gamma_0/\sqrt{n}$. The following results are highly inspired from [3], [2] and [75] with the slight technical differences that: the iterates are not bounded since no projection is used, the stochastic gradients are not almost surely bounded and a decreasing step-size is considered.

- Lemma 8 we give second and fourth moment bounds on $\|\theta_n - \theta^*\|$.
- Lemma 9 we give first and second moment bounds on $f(\bar{\theta}_n) - f(\theta^*)$.
- Lemma 10 we give first and second moment bounds on $f(\theta_n) - f(\theta^*)$.

We start by providing second and fourth moment bounds on $\|\theta_n - \theta^*\|$ in the following lemma.

**Lemma 8.** *Let (A.1, A.2, A.3, A.4) hold and consider the SGD iterates following Eq. (2). Assume $\gamma_n = \frac{\gamma_0}{\sqrt{n}}$. Then:*

$$\mathbb{E}\left[\|\theta_n - \theta^*\|^2\right] \leq \left(\|\theta_0 - \theta^*\|^2 + \gamma_0^2 R^2 \ln(en)\right) := C_n,$$

$$\mathbb{E}\left[\|\theta_n - \theta^*\|^4\right] \leq \left(\|\theta_0 - \theta^*\|^2 + 4\gamma_0^2 R^2 \ln(en)\right)^2 := D_n.$$

*Proof.* Starting from the definition of the SGD recursion eq. (2) we have:

$$\theta_n = \theta_{n-1} - \gamma_n f_n'(\theta_{n-1}), \tag{3}$$

and get the classical recursion:

$$\|\theta_n - \theta^*\|^2 = \|\theta_{n-1} - \theta^*\|^2 - 2\gamma_n \langle f_n'(\theta_{n-1}), \theta_{n-1} - \theta^*\rangle + \gamma_n^2 \|f_n'(\theta_{n-1})\|^2. \tag{4}$$

**Second moment bound.** We take the conditional expectation w.r.t the filtration $\mathcal{F}_{n-1} = \sigma((x_i, y_i)_{1 \leq i \leq n-1})$:

$$\mathbb{E}\left[\|\theta_n - \theta^*\|^2 \,|\mathcal{F}_{n-1}\right] = \|\theta_{n-1} - \theta^*\|^2 - 2\gamma_n \langle f'(\theta_{n-1}), \theta_{n-1} - \theta^*\rangle + \gamma_n^2 \|f_n'(\theta_{n-1})\|^2,$$

taking the full expectation and using that by convexity of $f$, $\langle f'(\theta_{n-1}), \theta_{n-1} - \theta^*\rangle \geq 0$, we obtain:

$$\mathbb{E}\left[\|\theta_n - \theta^*\|^2\right] \leq \mathbb{E}\left[\|\theta_{n-1} - \theta^*\|^2\right] + \gamma_n^2 R^2 \leq \|\theta_0 - \theta^*\|^2 + \gamma_0^2 R^2 \sum_{k=1}^{n} k^{-1} \leq \|\theta_0 - \theta^*\|^2 + \gamma_0^2 R^2 \ln(en).$$

**Fourth moment bound.** For the fourth-order moment bound, we take the square of Eq. (4):

$$\|\theta_n - \theta^*\|^4 = \|\theta_{n-1} - \theta^*\|^4 + 4\gamma_n^2 \langle f_n'(\theta_{n-1}), \theta_{n-1} - \theta^*\rangle^2 + \gamma_n^4 \|f_n'(\theta_{n-1})\|^4$$
$$- 4\gamma_n \langle f_n'(\theta_{n-1}), \theta_{n-1} - \theta^*\rangle \|\theta_{n-1} - \theta^*\|^2 - 4\gamma_n^3 \langle f_n'(\theta_{n-1}), \theta_{n-1} - \theta^*\rangle \|f_n'(\theta_{n-1})\|^2$$
$$+ 2\gamma_n^2 \|\theta_{n-1} - \theta^*\|^2 \|f_n'(\theta_{n-1})\|^2.$$

Taking the conditional expectation:

$$\mathbb{E}\left[\|\theta_n - \theta^*\|^4 \,|\mathcal{F}_{n-1}\right] = \|\theta_{n-1} - \theta^*\|^4 + 4\gamma_n^2 \langle \mathbb{E}\left[f_n'(\theta_{n-1}), \theta_{n-1} - \theta^*\rangle^2 |\mathcal{F}_n\right] + \gamma_n^4 \mathbb{E}\left[\|f_n'(\theta_{n-1})\|^4 \,|\mathcal{F}_n\right]$$
$$- 4\gamma_n \langle f'(\theta_{n-1}), \theta_{n-1} - \theta^*\rangle \|\theta_{n-1} - \theta^*\|^2 - 4\gamma_n^3 \mathbb{E}\left[\langle f_n'(\theta_{n-1}), \theta_{n-1} - \theta^*\rangle \|f_n'(\theta_{n-1})\|^2 \,|\mathcal{F}_n\right]$$
$$+ 2\gamma_n^2 \|\theta_{n-1} - \theta^*\|^2 \mathbb{E}\left[\|f_n'(\theta_{n-1})\|^2 \,|\mathcal{F}_n\right]$$
$$\leq \|\theta_{n-1} - \theta^*\|^4 + 6\gamma_n^2 R^2 \|\theta_{n-1} - \theta^*\|^2 + 3\gamma_n^4 (R^2)^2 + 2\gamma_n^2 \|\theta_{n-1} - \theta^*\|^2 R^2.$$

Taking the full expectation yields to

$$
\begin{aligned}
\mathbb{E}\left[\|\theta_n - \theta^*\|^4\right] \leq & \mathbb{E}\left[\|\theta_{n-1} - \theta^*\|^4\right] + 8\gamma_n^2 R^2 \mathbb{E}\left[\|\theta_{n-1} - \theta^*\|^2\right] + 3\gamma_n^4 (R^2)^2 \\
\leq & \mathbb{E}\left[\|\theta_{n-1} - \theta^*\|^4\right] + 8\gamma_n^2 R^2 D_{n-1} + 3\gamma_n^4 (R^2)^2 \\
\leq & \|\theta_0 - \theta^*\|^4 + 8\gamma_0^2 R^2 \sum_{k=1}^{n} \frac{\|\theta_0 - \theta^*\|^2 + \gamma_0^2 R^2 \ln(e(k-1))}{k} + 3\gamma_0^4 (R^2)^2 \sum_{k=1}^{n} \frac{1}{k^2} \\
\leq & \|\theta_0 - \theta^*\|^4 + 8\gamma_0^2 R^2 (\|\theta_0 - \theta^*\|^2 + \gamma_0^2 R^2 \ln(en)) \ln(en) + 3\gamma_0^4 (R^2)^2 \pi^2/6 \\
\leq & \|\theta_0 - \theta^*\|^4 + 8\gamma_0^2 \ln(en) R^2 \|\theta_0 - \theta^*\|^2 + \gamma_0^4 \ln(en)(R^2)^2 (8\ln(en) + \pi^2/3) \\
\leq & \left(\|\theta_0 - \theta^*\|^2 + 4\gamma_0^2 R^2 \ln(en)\right)^2.
\end{aligned}
$$

$\square$

We then give first and second moment bounds on the function value evaluated in the averaged iterate: $f(\bar{\theta}_n) - f(\theta^*)$.

**Lemma 9.** *Let (A.1, A.2, A.3, A.4) hold and consider the SGD iterates following Eq. (2). Assume $\gamma_n = \frac{\gamma_0}{\sqrt{n}}$. Then:*

$$
\mathbb{E}\left[f(\bar{\theta}_n)\right] - f(\theta^*) \leq \frac{1}{n} \sum_{k=0}^{n-1} \mathbb{E}\left[\langle f'(\theta_k), \theta_k - \theta^* \rangle\right] \leq \frac{1}{\sqrt{n}} \left[\frac{\|\theta_0 - \theta^*\|^2}{\gamma_0} + 2\gamma_0 R^2 \ln(en)\right],
$$

$$
\mathbb{E}\left[\left(f(\bar{\theta}_n) - f(\theta^*)\right)^2\right] \leq \mathbb{E}\left[\left(\frac{1}{n} \sum_{k=0}^{n-1} \langle f'(\theta_k), \theta_k - \theta^* \rangle\right)^2\right] \leq \frac{\ln(en)}{n} \left[\frac{\|\theta_0 - \theta^*\|^2}{\gamma_0} + 6\gamma_0 R^2 \ln(en)\right]^2.
$$

*Proof.* Rearranging Eq. (4) we have:

$$
2\langle f'(\theta_{n-1}), \theta_{n-1} - \theta^* \rangle = \gamma_n^{-1} \|\theta_{n-1} - \theta^*\|^2 - \gamma_n^{-1} \|\theta_n - \theta^*\|^2 + \gamma_n N_n + M_n,
$$

where we denote by $N_n := \|f'_n(\theta_{n-1})\|^2$ and $M_n := 2\langle f'(\theta_{n-1}) - f'_n(\theta_{n-1}), \theta_{n-1} - \theta^* \rangle$ which both satisfy $\mathbb{E}[N_n] \leq R^2$, $\mathbb{E}[M_n] = 0$ and $\mathbb{E}\left[M_n^2\right] \leq 8\mathbb{E}\left[\|\theta_{n-1} - \theta^*\|^2\right] R^2$.

Taking the sum of the previous equality for $k = 1$ to $k = n$, we obtain:

$$
2\sum_{k=0}^{n-1} \langle f'(\theta_k), \theta_k - \theta^* \rangle = \gamma_0^{-1} \|\theta_0 - \theta^*\|^2 - \gamma_n^{-1} \|\theta_n - \theta^*\|^2
$$

$$
+ \sum_{k=1}^{n-1} \|\theta_k - \theta^*\|^2 (\gamma_{k+1}^{-1} - \gamma_k^{-1}) + \sum_{k=1}^{n} (\gamma_k N_k + M_k). \quad (5)
$$

**First moment bound.**    The first result is obtained by directly taking the expectation, using Lemma 8 to bound $\mathbb{E}\left[\|\theta_k - \theta^*\|^2\right]$ and using the classical inequality $\sum_{k=1}^{n}\sqrt{k}^{-1} \leq 2\sqrt{n}$:

$$2\sum_{k=0}^{n-1}\mathbb{E}\left[\langle f'(\theta_k), \theta_k - \theta^*\rangle\right] = \gamma_0^{-1}\|\theta_0 - \theta^*\|^2 - \gamma_n^{-1}\mathbb{E}\left[\|\theta_n - \theta^*\|^2\right] + \sum_{k=1}^{n-1}\mathbb{E}\left[\|\theta_k - \theta^*\|^2\right](\gamma_{k+1}^{-1} - \gamma_k^{-1})$$

$$+ \sum_{k=1}^{n}\gamma_k\mathbb{E}\left[N_k\right]$$

$$\leq \gamma_0^{-1}\|\theta_0 - \theta^*\|^2 - \gamma_n^{-1}\mathbb{E}\left[\|\theta_n - \theta^*\|^2\right] + C_{n-1}\sum_{k=1}^{n-1}(\gamma_{k+1}^{-1} - \gamma_k^{-1}) + R^2\gamma_0\sum_{k=1}^{n}\sqrt{k}^{-1}$$

$$\leq \gamma_0^{-1}\|\theta_0 - \theta^*\|^2 + \gamma_n^{-1}C_{n-1} + 2R^2\gamma_0\sqrt{n}$$

$$\leq \gamma_0^{-1}\|\theta_0 - \theta^*\|^2 + \gamma_n^{-1}\left(\|\theta_0 - \theta^*\|^2 + \gamma_0^2 R^2\ln(en)\right) + 2R^2\gamma_0\sqrt{n}$$

$$\leq \sqrt{n}\left[\frac{2\|\theta_0 - \theta^*\|^2}{\gamma_0} + 4\gamma_0 R^2\ln(en)\right].$$

**Second moment bound.**    Notice that $\gamma_{k+1}^{-1} - \gamma_k^{-1} = \frac{1}{\gamma_0(\sqrt{k+1}+\sqrt{k})} \leq 1/(2\gamma_0\sqrt{k})$. To obtain the second-moment bound, we define $A_n := \gamma_0^{-1}\|\theta_0 - \theta^*\|^2 + \sum_{k=1}^{n-1}\frac{\|\theta_k-\theta^*\|^2}{2\gamma_0\sqrt{k}} + \sum_{k=1}^{n}(\gamma_k N_k + M_k)$ which satisfies the recursion formula for $A_0 = \frac{\|\theta_0-\theta^*\|^2}{2\gamma_0}$:

$$A_n = A_{n-1} + \frac{\|\theta_{n-1} - \theta^*\|^2}{2\gamma_0\sqrt{n}} + (\gamma_n N_n + M_n).$$

When proving the first moment bound we showed by induction that $\mathbb{E}\left[A_n\right] \leq \sqrt{n}[\frac{2\|\theta_0-\theta^*\|^2}{\gamma_0} + 4\gamma_0 R^2\ln(en)]$, hence:

$$\mathbb{E}\left[A_n^2\right] = \mathbb{E}\left[A_{n-1}^2\right] + \mathbb{E}\left[\frac{\|\theta_{n-1} - \theta^*\|^2}{2\gamma_0\sqrt{n}} + \gamma_n N_n + M_n\right]^2 + 2\mathbb{E}\left[A_{n-1}\frac{\|\theta_{n-1} - \theta^*\|^2}{2\gamma_0\sqrt{n}}\right]$$

$$+ 2\gamma_n\mathbb{E}\left[A_{n-1}N_n\right] + 2\mathbb{E}\left[A_{n-1}M_n\right]$$

$$\leq (1 + \frac{1}{n})\mathbb{E}\left[A_{n-1}^2\right] + \frac{D_{n-1}}{4\gamma_0^2} + 2\gamma_n R^2\mathbb{E}\left[A_n\right] + 3\frac{D_{n-1}}{4\gamma_0^2 n} + 3\gamma_n^2(R^2)^2 + 12C_{n-1}R^2,$$

since

$$\mathbb{E}\left[\frac{\|\theta_{n-1} - \theta^*\|^2}{2\gamma_0\sqrt{n}} + \gamma_n N_n + M_n\right]^2 \leq 3\frac{D_{n-1}}{4\gamma_0^2 n} + 3\gamma_n^2(R^2)^2 + 12C_{n-1}R^2.$$

Thus we obtain

$$\frac{\mathbb{E}\left[A_n^2\right]}{n+1} \leq \frac{\mathbb{E}\left[A_{n-1}^2\right]}{n} + \frac{D_{n-1}}{4\gamma_0^2} + 2\gamma_n R^2\mathbb{E}\left[A_n\right] + 3\frac{D_{n-1}}{4n\gamma_0^2} + 3\gamma_n^2(R^2)^2 + 12C_{n-1}R^2,$$

and we have then

$$\frac{\mathbb{E}\left[A_n^2\right]}{n+1} \leq \frac{\mathbb{E}\left[A_{n-1}^2\right]}{n} + \frac{[\|\theta_0 - \theta^*\|^2 + 11\gamma_0^2 R^2\ln(en)]^2}{\gamma_0^2(n+1)}.$$

Thus we find that

$$\mathbb{E}\left[A_n^2\right] \leq \frac{(n+1)\|\theta_0 - \theta^*\|^4/4 + (n+1)[\|\theta_0 - \theta^*\|^2 + 11\gamma_0^2 R^2\ln(en)]^2\ln(en)}{\gamma_0^2}.$$

Dividing by $n^2$ concludes the proof. □

In the following lemma we give a first and second moment bound on $f(\theta_n) - f(\theta^*)$. To do so we adapt the proof of [75].

**Lemma 10.** *Let (A.1, A.2, A.3, A.4) hold and consider the SGD iterates following Eq. (2). Assume* $\gamma_n = \frac{\gamma_0}{\sqrt{n}}$. *Then:*

$$\mathbb{E}\left[f(\theta_n)\right] - f(\theta^*) \leq \frac{\ln(en)}{\sqrt{n}} \left[ \frac{3\left\|\theta_0 - \theta^*\right\|^2}{\gamma_0} + 4R^2\gamma_0 \ln(en) \right],$$

$$\mathbb{E}\left[(f(\theta_n) - f(\theta^*))^2\right] \leq \frac{8\ln^2 en}{n} \left[ 4\frac{\left\|\theta_0 - \theta^*\right\|^2}{\gamma_0} + 20\gamma_0 R^2 \ln en \right]^2.$$

*Proof.* We adapt the proof of [75]. We note that from eq. (3) and for any $\theta \in \mathbb{R}^d$:

$$\left\|\theta_n - \theta\right\|^2 = \left\|\theta_{n-1} - \theta\right\|^2 - 2\gamma_n \langle f'_n(\theta_{n-1}), \theta_{n-1} - \theta \rangle + \gamma_n^2 \left\|f'_n(\theta_{n-1})\right\|^2. \tag{6}$$

Rearranging Eq. (6) we have:

$$2\langle f'(\theta_{n-1}), \theta_{n-1} - \theta \rangle = \gamma_n^{-1} \left\|\theta_{n-1} - \theta\right\|^2 - \gamma_n^{-1} \left\|\theta_n - \theta\right\|^2 + \gamma_n N_n + M_n^\theta,$$

where we denote by $N_n := \left\|f'_n(\theta_{n-1})\right\|^2$ and $M_n^\theta := 2\langle f'(\theta_{n-1}) - f'_n(\theta_{n-1}), \theta_{n-1} - \theta \rangle$. Note that they satisfy $\mathbb{E}\left[N_n\right] \leq R^2$, $\mathbb{E}\left[M_n^\theta\right] = 0$ and $\mathbb{E}\left[(M_n^\theta)^2\right] \leq 8\mathbb{E}\left[\left\|\theta_{n-1} - \theta\right\|^2\right] R^2$.

Therefore summing from $k = n + 1 - t$ to $k = n$, and applying for $\theta = \theta_{n-t}$ we have:

$$2\sum_{k=n-t}^{n-1} \langle f'(\theta_k), \theta_k - \theta_{n-t} \rangle \leq \sum_{k=n+1-t}^{n-1} \left\|\theta_k - \theta_{n-t}\right\|^2 (\gamma_{k+1}^{-1} - \gamma_k^{-1}) + \sum_{k=n+1-t}^{n} \gamma_k N_k + \sum_{k=n+1-t}^{n} M_k^{n-t}$$
$$:= B_{n-t}^n,$$

where we write $M_k^{n-t} = M_k^{\theta_{n-t}}$. Using the fact that $f$ is convex we get that

$$2\sum_{k=n-t}^{n-1} (f(\theta_k) - f(\theta_{n-t})) \leq B_{n-t}^n. \tag{7}$$

As in the proof of [75], let $S_t = \frac{1}{t} \sum_{k=n-t}^{n-1} f(\theta_k)$ be the average value of the last $t$ iterates. Rewriting eq. (7) we get:

$$-f(\theta_{n-t}) \leq -S_t + \frac{B_{n-t}^n}{2t}.$$

The trick is to note that:

$$(t-1)S_{t-1} = tS_t - f(\theta_{n-t}) \leq tS_t - S_t + \frac{B_{n-t}^n}{2t},$$

Dividing by $t - 1$ we immediately obtain:

$$S_{t-1} \leq S_t + \frac{B_{n-t}^n}{2t(t-1)}.$$

Summing from $t = 2$ to $t = n$:

$$f(\theta_{n-1}) = S_1 \leq S_n + \sum_{t=2}^{n} \frac{B_{n-t}^n}{2t(t-1)}. \tag{8}$$

**First moment bound.** We obtain the first moment bound by taking the expectation of eq. (8):

$$\mathbb{E}\left[f(\theta_{n-1})\right] - f(\theta^*) \leq \mathbb{E}\left[S_n\right] - f(\theta^*) + \sum_{t=2}^{n} \frac{\mathbb{E}\left[B_{n-t}^n\right]}{2t(t-1)}.$$

With

$$\mathbb{E}\left[B_{n-t}^n\right] \le \sum_{k=n+1-t}^{n-1} \mathbb{E}\left[\|\theta_k - \theta_{n-t}\|^2\right](\gamma_{k+1}^{-1} - \gamma_k^{-1}) + \sum_{k=n+1-t}^{n}\left(\gamma_k\mathbb{E}\left[N_k\right] + \mathbb{E}\left[M_k^{n-t}\right]\right)$$

$$\le 4C_{n-1}(\gamma_n^{-1} - \gamma_{n+1-t}^{-1}) + R^2\sum_{k=n+1-t}^{n}\gamma_k$$

$$\le [4C_{n-1}/\gamma_0 + 2R^2\gamma_0](\sqrt{n} - \sqrt{n-t})$$

$$\le [4C_{n-1}/\gamma_0 + 2R^2\gamma_0]\frac{t}{\sqrt{n} + \sqrt{n-t}} \le [4C_{n-1}/\gamma_0 + 2R^2\gamma_0]\frac{t}{\sqrt{n}},$$

where we have used that by integration by part $\sum_{k=n+1-t}^{n} 1/\sqrt{k} = 2(\sqrt{n} - \sqrt{n-t})$. Thus

$$\mathbb{E}\left[f(\theta_{n-1})\right] - f(\theta^*) \le \mathbb{E}\left[S_n\right] - f(\theta^*) + \frac{4C_{n-1}/\gamma_0 + 2R^2\gamma_0}{2\sqrt{n}} \le \mathbb{E}\left[S_n\right] - f(\theta^*) + \frac{4C_{n-1}/\gamma_0 + 2R^2\gamma_0}{2\sqrt{n}}\ln(en).$$

We can now bound $C_{n-1}$ using Lemma 8 and $\mathbb{E}\left[S_n\right] - f(\theta^*)$ using Lemma 9 to obtain:

$$\mathbb{E}\left[f(\theta_{n-1})\right] - f(\theta^*) \le \frac{1}{\sqrt{n}}\left[\frac{\|\theta_0 - \theta^*\|^2}{2\gamma_0} + \gamma_0 R^2\ln(en)\right] + \frac{4C_{n-1}/\gamma_0 + 2R^2\gamma_0}{2\sqrt{n}}\ln(en)$$

$$\le \frac{1}{\sqrt{n}}\left[\frac{\|\theta_0 - \theta^*\|^2}{2\gamma_0} + \gamma_0 R^2\ln(en)\right] + \frac{4\|\theta_0 - \theta^*\|^2/\gamma_0 + 6R^2\gamma_0\ln(en)}{2\sqrt{n}}\ln(en)$$

$$\le \left(3\|\theta_0 - \theta^*\|^2/\gamma_0 + 4R^2\gamma_0\ln(en)\right)\frac{\ln(en)}{\sqrt{n}}.$$

**Second moment bound.** For the second moment bound, we obtain taking the square in both sides of Eq. (8):

$$\mathbb{E}\left[(f(\theta_{n-1}) - f(\theta^*))^2\right] \le 2\mathbb{E}\left[(S_n - f(\theta^*))^2\right] + 2\mathbb{E}\left[\left(\sum_{t=2}^{n}\frac{B_{n-t}^n}{2t(t-1)}\right)^2\right]. \tag{9}$$

We can bound the first term using the second bound of Lemma 9:

$$\mathbb{E}\left[(S_n - f(\theta^*))^2\right] \le \mathbb{E}\left[\left(\frac{1}{n}\sum_{k=0}^{n-1}\langle f'(\theta_k), \theta_k - \theta^*\rangle\right)^2\right] \le \frac{\ln(en)}{n}\left[\frac{\|\theta_0 - \theta^*\|^2}{\gamma_0} + 6\gamma_0 R^2\ln(en)\right]^2.$$

For the second term we compute:

$$\sum_{t=2}^{n}\frac{B_{n-t}^n}{2t(t-1)} = \sum_{t=2}^{n}\sum_{k=n+1-t}^{n-1}\frac{\|\theta_k - \theta_{n-t}\|^2(\gamma_{k+1}^{-1} - \gamma_k^{-1})}{2t(t-1)} + \sum_{t=2}^{n}\sum_{k=n+1-t}^{n}\frac{\gamma_k N_k}{2t(t-1)} + \sum_{t=2}^{n}\sum_{k=n+1-t}^{n}\frac{M_k^{n-t}}{2t(t-1)},$$

and individually bound:

$$\mathbb{E}\left[\left(\sum_{t=2}^{n}\frac{B_{n-t}^n}{2t(t-1)}\right)^2\right] \le 3\mathbb{E}\left[\left(\sum_{t=2}^{n}\sum_{k=n+1-t}^{n-1}\frac{\|\theta_k - \theta_{n-t}\|^2(\gamma_{k+1}^{-1} - \gamma_k^{-1})}{2t(t-1)}\right)^2\right]$$

$$+ 3\mathbb{E}\left[\left(\sum_{t=2}^{n}\sum_{k=n+1-t}^{n}\frac{\gamma_k N_k}{2t(t-1)}\right)^2\right] + 3\mathbb{E}\left[\left(\sum_{t=2}^{n}\sum_{k=n+1-t}^{n}\frac{M_k^{n-t}}{2t(t-1)}\right)^2\right]. \tag{10}$$

For the first term, we use that for $1 \le i, j \le n$:

$$\mathbb{E}\left[\|\theta_i - \theta_j\|^4\right] \le 8\mathbb{E}\left[\|\theta_i - \theta_*\|^4\right] + 8\mathbb{E}\left[\|\theta_j - \theta_*\|^4\right] \le 16D_n.$$

Therefore we use the Minkowski inequality ( $\sqrt{\mathbb{E}\left[(a+b)^2\right]} \leq \sqrt{\mathbb{E}\left[a^2\right]} + \sqrt{\mathbb{E}\left[b^2\right]}$ ) to obtain:

$$\mathbb{E}\left[\left(\sum_{t=2}^{n}\sum_{k=n+1-t}^{n-1}\frac{\|\theta_k - \theta_{n-t}\|^2\left(\gamma_{k+1}^{-1} - \gamma_k^{-1}\right)}{2t(t-1)}\right)^2\right] \leq \left(\sum_{t=2}^{n}\sum_{k=n+1-t}^{n-1}\frac{\gamma_{k+1}^{-1} - \gamma_k^{-1}}{2t(t-1)}\sqrt{\mathbb{E}\left[\|\theta_k - \theta_{n-t}\|^4\right]}\right)^2$$

$$\leq \frac{16 D_n}{4n\gamma_0^2}\left[\sum_{t=2}^{n}\frac{t-1}{t(t-1)}\right]^2 \leq \frac{4 D_n}{n\gamma_0^2}\ln^2(en).$$

Hence using Lemma 8:

$$\mathbb{E}\left[\left(\sum_{t=2}^{n}\sum_{k=n+1-t}^{n-1}\frac{\|\theta_k - \theta_{n-t}\|^2\left(\gamma_{k+1}^{-1} - \gamma_k^{-1}\right)}{2t(t-1)}\right)^2\right] \leq \frac{4\ln^2(en)}{n\gamma_0^2}\left(\|\theta_0 - \theta^*\|^2 + 4\gamma_0^2 R^2 \ln(en)\right)^2.$$

$$(11)$$

For the second term, we proceed in the same way. A classical result on the fourth moment of a Gaussian random variable gives: $\mathbb{E}\left[N_k^2\right] = \mathbb{E}\left[\|x\|_2^4\right] \leq 3\mathbb{E}\left[\|x\|_2^2\right]^2 \leq 3R^4$. Hence:

$$\mathbb{E}\left[\left(\sum_{t=2}^{n}\sum_{k=n+1-t}^{n}\frac{\gamma_k N_k}{2t(t-1)}\right)^2\right] \leq \left(\sum_{t=2}^{n}\sum_{k=n+1-t}^{n}\frac{\gamma_k\sqrt{\mathbb{E}\left[N_k^2\right]}}{2t(t-1)}\right)^2$$

$$\leq \left(\gamma_0\sqrt{3R^4}\sum_{t=2}^{n}\frac{1}{2t(t-1)}\sum_{k=n+1-t}^{n}\sqrt{k}^{-1}\right)^2$$

$$\leq \left(\gamma_0\sqrt{3R^4}\sum_{t=2}^{n}\frac{\sqrt{n} - \sqrt{n-t}}{t(t-1)}\right)^2$$

$$\leq \left(\frac{\gamma_0\sqrt{3R^4}}{2\sqrt{n}}\sum_{t=2}^{n}\frac{1}{(t-1)}\right)^2 \leq \frac{3\gamma_0^2 R^4 \ln^2(en)}{4n}. \qquad (12)$$

For the third term, denoting:

$$M_k^{n-t} = 2\langle f'(\theta_{k-1}) - f'_k(\theta_{k-1}), \theta_{k-1} - \theta_{n-t}\rangle$$
$$:= 2\langle \zeta_k, \theta_{k-1} - \theta_{n-t}\rangle.$$

Let $\alpha_t = \frac{1}{t(t-1)}$ and $\Delta_n^k = \sum_{t=n+1-k}^{n} \alpha_t = \frac{1}{n-k} - \frac{1}{n}$. Using martingale second moment expansions yields:

$$\mathbb{E}\left[\left(\sum_{t=2}^{n} \sum_{k=n+1-t}^{n} \frac{M_k^{n-t}}{2t(t-1)}\right)^2\right] = \mathbb{E}\left[\left(\sum_{k=1}^{n-1} \left\langle \zeta_k, \sum_{t=n+1-k}^{n} \frac{\theta_{k-1}-\theta_{n-t}}{t(t-1)}\right\rangle\right)^2\right]$$

$$= \sum_{k=1}^{n-1} \mathbb{E}\left[\left\langle \zeta_k, \sum_{t=n+1-k}^{n} \frac{\theta_{k-1}-\theta_{n-t}}{t(t-1)}\right\rangle^2\right]$$

$$\leq 2R^2 \sum_{k=1}^{n-1} \mathbb{E}\left[\left\|\sum_{t=n+1-k}^{n} \frac{\theta_{k-1}-\theta_{n-t}}{t(t-1)}\right\|^2\right]$$

$$\leq 2R^2 \sum_{k=1}^{n-1} (\Delta_n^k)^2 \mathbb{E}\left[\left\|\sum_{t=n+1-k}^{n} \frac{\alpha_t}{\Delta_n^k}(\theta_{k-1}-\theta_{n-t})\right\|^2\right]$$

$$\leq 2R^2 \sum_{k=1}^{n-1} (\Delta_n^k)^2 \sum_{t=n+1-k}^{n} \frac{\alpha_t}{\Delta_n^k} \mathbb{E}\left[\|\theta_{k-1}-\theta_{n-t}\|^2\right]$$

$$\leq 2R^2 \sum_{k=1}^{n-1} \Delta_n^k \sum_{t=n+1-k}^{n} \alpha_t \mathbb{E}\left[\|\theta_{k-1}-\theta_{n-t}\|^2\right]$$

$$\leq 2R^2 \sum_{k=1}^{n-1} [(n-k)^{-1} - n^{-1}] \sum_{t=n+1-k}^{n} \frac{\mathbb{E}\left[\|\theta_{k-1}-\theta_{n-t}\|^2\right]}{t(t-1)}.$$

Notice that taking the expectation in eq. (6), using $f$'s convexity and the fact that the stochastic gradients are bounded in expectation:

$$\mathbb{E}\left[\|\theta_i - \theta\|^2\right] \leq \mathbb{E}\left[\|\theta_{i-1} - \theta\|^2\right] - 2\gamma_i(f(\theta_i) - f(\theta)) + \gamma_i^2 R^2,$$

Hence summing form $i = n - t + 1$ to $i = k - 1$:

$$\mathbb{E}\left[\|\theta_{k-1} - \theta\|^2\right] \leq \mathbb{E}\left[\|\theta_{n-t} - \theta\|^2\right] - 2\gamma_0 \sum_{i=n-t+1}^{k-1} \frac{\mathbb{E}\left[\langle f(\theta_{i-1}), \theta_{i-1} - \theta\rangle\right]}{\sqrt{i}} + \gamma_0^2 R^2 \sum_{i=n-t+1}^{k-1} \frac{1}{i},$$

This leads to, if $n - t \geq 1$

$$\mathbb{E}\left[\|\theta_{k-1} - \theta_{n-t}\|^2\right] \leq \gamma_0^2 R^2 [\ln(k-1) - \ln(n-t)] + 2\gamma_0 \sum_{i=n-t}^{k-2} \frac{\mathbb{E}\left[f(\theta_{n-t}) - f(\theta_i)\right]}{\sqrt{i+1}},$$

and if $n - t = 0$, with the convention $\ln 0 = 0$:

$$\mathbb{E}\left[\|\theta_{k-1} - \theta_0\|^2\right] \leq \gamma_0^2 R^2 [\ln(e(k-1))] + 2\gamma_0 \sum_{i=0}^{k-2} \frac{\mathbb{E}\left[f(\theta_0) - f(\theta_i)\right]}{\sqrt{i+1}}.$$

Hence:

$$\sum_{k=1}^{n-1} [(n-k)^{-1} - n^{-1}] \sum_{t=n+1-k}^{n} \frac{\mathbb{E}\left[\|\theta_{k-1} - \theta_{n-t}\|^2\right]}{t(t-1)}$$

$$\leq \gamma_0^2 R^2 \sum_{k=1}^{n-1} [(n-k)^{-1} - n^{-1}] \left[\frac{\ln e(k-1)}{n(n-1)} + \sum_{t=n+1-k}^{n-1} \frac{\ln(k-1) - \ln(n-t)}{t(t-1)}\right]$$

$$+ 2\gamma_0 \sum_{k=1}^{n-1} [(n-k)^{-1} - n^{-1}] \sum_{t=n+1-k}^{n} \frac{1}{t(t-1)} \sum_{i=n-t}^{k-2} \frac{\mathbb{E}\left[f(\theta_{n-t}) - f(\theta_i)\right]}{\sqrt{i+1}}.$$

The function $x \mapsto -\frac{\ln(1-x)}{x}$ is increasing on $[0,1]$. Hence for all $2 \le t \le n-1$: $-\frac{\ln(1-\frac{t}{n})}{\frac{t}{n}} \le \ln(n)\frac{n}{n-1} \le 2\ln(n)$. Hence we can upper-bound:

$$\sum_{k=1}^{n-1}[(n-k)^{-1}-n^{-1}]\left[\frac{\ln e(k-1)}{n(n-1)} + \sum_{t=n+1-k}^{n-1}\frac{\ln(k-1)-\ln(n-t)}{t(t-1)}\right]$$

$$\le \sum_{k=1}^{n-1}(n-k)^{-1}\frac{\ln e(k-1)}{n(n-1)} + \sum_{k=1}^{n-1}(n-k)^{-1}\left[\sum_{t=2}^{n-1}\frac{\ln(n)-\ln(n-t)}{t(t-1)}\right]$$

$$= \frac{\ln en}{n} + \frac{1}{n}\sum_{k=1}^{n}k^{-1}\sum_{t=2}^{n-1}-\frac{\ln(1-\frac{t}{n})}{\frac{t}{n}(t-1)}$$

$$\le \frac{\ln en}{n} + \frac{2\ln n}{n}\sum_{k=1}^{n}k^{-1}\sum_{t=2}^{n-1}\frac{1}{t-1}$$

$$\le \frac{3\ln^3 en}{n}.$$

For the second term, let $A_n = \left(\frac{3\|\theta_0-\theta^*\|^2}{\gamma_0} + 4R^2\gamma_0\ln(en)\right)\ln(en)$, according to Lemma 10, for $n-t > 0$, $\mathbb{E}\left[f(\theta_{n-t})-f(\theta^*)\right] \le A_{n-t}\frac{1}{\sqrt{n-t}} \le A_n\frac{1}{\sqrt{n-t}}$. Furthermore, notice that rearranging eq. (4) we obtain $f(\theta_0) - f(\theta^*) \le \frac{\|\theta_0-\theta^*\|^2}{2\gamma_0} + \frac{\gamma_0 R^2}{2} \le A_1$. Hence:

$$\sum_{k=1}^{n-1}[(n-k)^{-1}-n^{-1}]\sum_{t=n+1-k}^{n}\frac{1}{t(t-1)}\sum_{i=n-t}^{k-2}\frac{\mathbb{E}\left[f(\theta_{n-t})-f(\theta^*)-(f(\theta_i)-f(\theta^*))\right]}{\sqrt{i+1}}$$

$$\le \sum_{k=1}^{n-1}(n-k)^{-1}\sum_{t=n+1-k}^{n}\frac{1}{t(t-1)}\sum_{i=n-t}^{k-2}\frac{\mathbb{E}\left[f(\theta_{n-t})-f(\theta^*)\right]}{\sqrt{i+1}}$$

$$\le A_n\sum_{k=1}^{n-1}(n-k)^{-1}\left[\sum_{t=n+1-k}^{n-1}\frac{1}{t(t-1)}\sum_{i=n-t}^{k-2}\frac{1}{\sqrt{i+1}}\frac{1}{\sqrt{n-t}} + \frac{1}{n(n-1)}\sum_{i=0}^{k-2}\frac{1}{\sqrt{i+1}}\right]$$

$$\le A_n\sum_{k=1}^{n-1}(n-k)^{-1}\sum_{t=2}^{n-1}\frac{1}{t(t-1)}\frac{1}{\sqrt{n-t}}\sum_{i=n-t}^{n}\frac{1}{\sqrt{i+1}}$$

$$\le A_n\sum_{k=1}^{n-1}(n-k)^{-1}\sum_{t=2}^{n-1}\frac{1}{t(t-1)}\left((1-\frac{t}{n})^{-1/2}-1\right)$$

$$\le 2A_n\frac{1}{n}\sum_{k=1}^{n-1}(n-k)^{-1}\left(\frac{1}{n}\sum_{t=2}^{n-1}\frac{1}{(\frac{t}{n})^2}\left((1-\frac{t}{n})^{-1/2}-1\right)\right)$$

$$\le 6A_n\frac{\ln en}{n}\sum_{k=1}^{n}(n-k)^{-1}$$

$$\le 6A_n\frac{\ln^2 en}{n},$$

where we have used Lemma 20 to upper bound the Riemann sum. Hence:

$$
\mathbb{E}\left[\left(\sum_{t=2}^{n}\sum_{k=n+1-t}^{n}\frac{M_k^{n-t}}{2t(t-1)}\right)^2\right] \leq 2R^2[\gamma_0^2 R^2 \frac{3\ln^3 en}{n} + 2\gamma_0 6A_n \frac{\ln^2 en}{n}]
$$

$$
\leq \frac{6R^2}{n}[\gamma_0^2 R^2 \ln^3 en + 4\gamma_0 A_n \ln^2 en]
$$

$$
\leq \frac{6R^2}{n}[\gamma_0^2 R^2 \ln^3 en + 12\|\theta_0 - \theta^*\|^2 \ln^2 en + 16R^2\gamma_0^2 \ln^4 en]
$$

$$
= \frac{6R^2 \ln^2 en}{n}[12\|\theta_0 - \theta^*\|^2 + 17R^2\gamma_0^2 \ln^2 en]. \tag{13}
$$

Injecting eqs. (11) to (13) into eq. (10) we get:

$$
\mathbb{E}\left[\left(\sum_{t=2}^{n}\frac{B_{n-t}^n}{2t(t-1)}\right)^2\right] \leq \frac{3}{n}\Big(\frac{4\ln^2(en)}{\gamma_0^2}\left(\|\theta_0 - \theta^*\|^2 + 4\gamma_0^2 R^2 \ln(en)\right)^2 + \frac{3\gamma_0^2 R^4 \ln^2(en)}{4}
$$

$$
+ 6R^2 \ln^2(en)[12\|\theta_0 - \theta^*\|^2 + 17R^2\gamma_0^2 \ln^2(en)]\Big)
$$

$$
\leq \frac{3\ln^2 en}{n}\left[4\frac{\|\theta_0 - \theta^*\|^2}{\gamma_0} + 20\gamma_0 R^2 \ln en\right]^2.
$$

Finally injecting this last inequality along with Lemma 9 in eq. (9) we obtain:

$$
\mathbb{E}\left[(f(\theta_{n-1}) - f(\theta^*))^2\right] \leq 2\frac{\ln(en)}{n}\left[\frac{\|\theta_0 - \theta^*\|^2}{\gamma_0} + 6\gamma_0 R^2 \ln(en)\right]^2 + \frac{6\ln^2 en}{n}\left[4\frac{\|\theta_0 - \theta^*\|^2}{\gamma_0} + 20\gamma_0 R^2 \ln en\right]^2
$$

$$
\leq \frac{8\ln^2 en}{n}\left[4\frac{\|\theta_0 - \theta^*\|^2}{\gamma_0} + 20\gamma_0 R^2 \ln en\right]^2.
$$

$\square$

## B  General results on the function $f$

In the following section we prove the general results on $f$ which are given Section 3. We also provide a few more results which will be useful for proving the main convergence guarantee result.

**Proof of Lemmas 1 to 3.**  Note that $f(\theta) = \mathbb{E}\left[\mathbb{E}\left[|\varepsilon + b - \langle x, \theta - \theta^*\rangle| \mid b\right]\right]$. Since $b$ is independent of $x$ and $\varepsilon$, given outlier $b$, $\varepsilon + b - \langle x, \theta - \theta^*\rangle$ is a random variable following $\mathcal{N}(b, \sigma^2 + \|\theta - \theta^*\|_H^2)$. Hence $|\varepsilon + b - \langle x, \theta - \theta^*\rangle|$ is a folded normal distribution and its expectation has a known closed form [53]:

$$
\mathbb{E}\left[|\varepsilon + b - \langle x, \theta - \theta^*\rangle| \mid b\right] = \sqrt{\frac{2}{\pi}}\sqrt{\sigma^2 + \|\theta - \theta^*\|_H^2} \exp\left(-\frac{b^2}{2(\sigma^2 + \|\theta - \theta^*\|_H^2)}\right)
$$

$$
+ b\,\mathrm{erf}\left(\frac{b}{\sqrt{2(\sigma^2 + \|\theta - \theta^*\|_H^2)}}\right).
$$

$f$'s closed form formula immediately follows by taking the expectation over the outlier distribution.

Note that in what follows the two successive differentations are valid since they lead to uniformly bounded functions that therefore have finite expectations. The first differentiation of $f$ leads to :

$$f'(\theta) = \mathbb{E}_b \left[ \sqrt{\frac{2}{\pi}} \frac{1}{\sqrt{\sigma^2 + \|\theta - \theta^*\|_H^2}} \exp\left(-\frac{b^2}{2(\sigma^2 + \|\theta - \theta^*\|_H^2)}\right) H(\theta - \theta^*) \right.$$

$$+ \sqrt{\frac{1}{2\pi}} \exp\left(-\frac{b^2}{2(\sigma^2 + \|\theta - \theta^*\|_H^2)}\right) \frac{b}{(\sigma^2 + \|\theta - \theta^*\|_H^2)^{3/2}} H(\theta - \theta^*)$$

$$\left. - \sqrt{\frac{1}{2\pi}} \exp\left(-\frac{b^2}{2(\sigma^2 + \|\theta - \theta^*\|_H^2)}\right) \frac{o}{(\sigma^2 + \|\theta - \theta^*\|_H^2)^{3/2}} H(\theta - \theta^*) \right]$$

$$= \sqrt{\frac{2}{\pi}} \frac{1}{\sqrt{\sigma^2 + \|\theta - \theta^*\|_H^2}} \mathbb{E}_b \left[ \exp\left(-\frac{b^2}{2(\sigma^2 + \|\theta - \theta^*\|_H^2)}\right) \right] H(\theta - \theta^*),$$

which can be rewritten as $f'(\theta) = \alpha(\|\theta - \theta^*\|_H) H(\theta - \theta^*)$.

The second derivative of $f$ leads to:

$$f''(\theta) = \sqrt{\frac{2}{\pi}} \mathbb{E}_b \left[ \exp\left(-\frac{b^2}{2(\sigma^2 + \|\theta - \theta^*\|_H^2)}\right) \right.$$

$$\left. \left(-\frac{H(\theta - \theta^*)^{\otimes 2} H}{(\sigma^2 + \|\theta - \theta^*\|_H^2)^{3/2}} \left(1 - \frac{b^2}{2(\sigma^2 + \|\theta - \theta^*\|_H^2)}\right) + \frac{H}{\sqrt{\sigma^2 + \|\theta - \theta^*\|_H^2}}\right) \right].$$

Setting $\theta = \theta^*$ immediately leads to:

$$f''(\theta^*) = \sqrt{\frac{2}{\pi}} \frac{1}{\sigma} \mathbb{E}_b \left[ \exp\left(-\frac{b^2}{2\sigma^2}\right) \right] H$$

$$= \sqrt{\frac{2}{\pi}} \frac{1 - \tilde{\eta}}{\sigma} H.$$

This concludes the proof of Lemmas 1 to 3. $\qquad\square$

We now prove a few more results on $f$. The following lemma shows that $f(\theta) - f(\theta^*)$ and $\|\theta - \theta^*\|_H^2$ are closely related.

**Lemma 11.** *Let (A.1, A.2, A.3, A.4) hold. Then,*

*For $\|\theta - \theta^*\|_H \geq \sigma$:*

$$\|\theta - \theta^*\|_H^2 \leq \frac{10}{(1 - \tilde{\eta})^2} (f(\theta) - f(\theta^*))^2.$$

*For $\|\theta - \theta^*\|_H \leq \sigma$:*

$$\|\theta - \theta^*\|_H^2 \leq \frac{4\sigma}{1 - \tilde{\eta}} (f(\theta) - f(\theta^*)).$$

*Proof.* To prove these inequalities we set $b \in \mathbb{R}$ and we take the expectation over the outlier distribution afterwards.

Let $f_b(\theta) = b\,\text{erf}\left(\frac{b}{\sqrt{2(\sigma^2 + \|\theta - \theta^*\|_H^2)}}\right) + \sqrt{\frac{2}{\pi}} \sqrt{\sigma^2 + \|\theta - \theta^*\|_H^2} \exp\left(-\frac{b^2}{2(\|\theta - \theta^*\|_H^2 + \sigma^2)}\right)$. We render the analysis dimensionless by letting :

$$\tilde{f}_{\tilde{b}}(\tilde{\sigma}) = \tilde{b}\,\text{erf}\left(\frac{\tilde{b}}{\sqrt{1 + \tilde{\sigma}^2}}\right) + \frac{1}{\sqrt{\pi}} \sqrt{1 + \tilde{\sigma}^2} \exp\left(-\frac{\tilde{b}^2}{1 + \tilde{\sigma}^2}\right).$$

Therefore notice that $f_b(\theta) = \sqrt{2}\sigma \tilde{f}_{\tilde{b}}(\tilde{\sigma})$ where $\tilde{\sigma} = \frac{\|\theta - \theta^*\|_H}{\sigma}$ and $\tilde{b} = \frac{b}{\sqrt{2}\sigma}$.

**First inequality:**  We first show that for $\tilde{\sigma} \geq 1$, $\tilde{\sigma} \leq \frac{\sqrt{\pi}}{\sqrt{2}-1} \exp\left(\tilde{b}^2\right) (\tilde{f}_{\tilde{b}}(\tilde{\sigma}) - \tilde{f}_{\tilde{b}}(0))$. Indeed $\tilde{f}_{\tilde{b}}$ is convex (as for $f$ it can be seen as: $\mathbb{E}\left[\left|\varepsilon + \tilde{b} - x\tilde{\sigma}\right|\right]$ where $\varepsilon, x \sim \mathcal{N}(0,1)$ independent). Hence $\frac{\tilde{f}_{\tilde{b}}(\tilde{\sigma}) - \tilde{f}_{\tilde{b}}(0)}{\tilde{\sigma}}$ is increasing, therefore for all $\tilde{\sigma} \geq 1$, $\frac{\tilde{f}_{\tilde{b}}(\tilde{\sigma}) - \tilde{f}_{\tilde{b}}(0)}{\tilde{\sigma}} \geq \tilde{f}_{\tilde{b}}(1) - \tilde{f}_{\tilde{b}}(0)$. Notice that using Lemma 18:

$$\tilde{f}_{\tilde{b}}(1) - \tilde{f}_{\tilde{b}}(0) = \tilde{b}\left(\operatorname{erf}\left(\frac{\tilde{b}}{\sqrt{2}}\right) - \operatorname{erf}\left(\tilde{b}\right)\right) + \sqrt{\frac{2}{\pi}}\exp\left(-\frac{\tilde{b}^2}{2}\right) - \frac{1}{\sqrt{\pi}}\exp\left(-\tilde{b}^2\right)$$

$$\geq \frac{\sqrt{2}-1}{\sqrt{\pi}}\exp\left(-\tilde{b}^2\right).$$

Hence for all $\tilde{\sigma} \geq 1$, $\tilde{\sigma}\frac{\sqrt{2}-1}{\sqrt{\pi}}\exp\left(-\tilde{b}^2\right) \leq (\tilde{f}_{\tilde{b}}(\tilde{\sigma}) - \tilde{f}_{\tilde{b}}(0))$. Now, for $\theta \in \mathbb{R}^d$ such that $\|\theta - \theta^*\|_H \geq \sigma$, let $\tilde{\sigma} = \frac{\|\theta - \theta^*\|_H}{\sigma} \geq 1$ and $\tilde{b} = \frac{b}{\sqrt{2}\sigma}$:

$$f_b(\theta) - f_b(\theta^*) = \sqrt{2}\sigma(\tilde{f}_{\tilde{b}}(\tilde{\sigma}) - \tilde{f}_{\tilde{b}}(0))$$

$$\geq \sqrt{2}\sigma\tilde{\sigma}\frac{\sqrt{2}-1}{\sqrt{\pi}}\exp\left(-\tilde{b}^2\right)$$

$$= \frac{\sqrt{2}(\sqrt{2}-1)}{\sqrt{\pi}}\|\theta - \theta^*\|_H \exp\left(-\frac{b^2}{2\sigma^2}\right).$$

Taking the expectation over $b$ we immediately get that for $\|\theta - \theta^*\|_H \geq \sigma$:

$$\|\theta - \theta^*\|_H \, \mathbb{E}_b\left[\exp\left(-\frac{b^2}{2\sigma^2}\right)\right] \leq \frac{\pi}{2(\sqrt{2}-1)^2}(f(\theta) - f(\theta^*)),$$

which leads to the first inequality since $\frac{\pi}{2(\sqrt{2}-1)^2} \leq 10$.

**Second inequality:**  The second inequality is shown the same way as for the first inequality. This time we use Lemma 19: for $\tilde{\sigma} \leq 1$, $\tilde{\sigma}^2 \leq 4\exp\left(\tilde{b}^2\right) (\tilde{f}_{\tilde{b}}(\tilde{\sigma}) - \tilde{f}_{\tilde{b}}(0))$. This leads to: for $\|\theta - \theta^*\|_H \leq \sigma$,

$$f_b(\theta) - f_b(\theta^*) = \sqrt{2}\sigma(\tilde{f}_{\tilde{b}}(\tilde{\sigma}) - \tilde{f}_{\tilde{b}}(0))$$

$$\leq \sqrt{2}\sigma\frac{\tilde{\sigma}^2}{5}\exp\left(-\tilde{b}^2\right)$$

$$= \sqrt{2}\frac{\|\theta - \theta^*\|_H^2}{5\sigma}\exp\left(-\frac{b^2}{2\sigma^2}\right).$$

$$\frac{\|\theta - \theta^*\|_H^2}{\sigma}\exp\left(-\frac{b^2}{2\sigma^2}\right) \leq \frac{5}{\sqrt{2}}(f_b(\theta) - f_b(\theta^*)).$$

Taking the expectation over $b$ concludes the proof. $\qquad\square$

The following inequality upper-bounds the classical prediction loss $\mathbb{E}\left[\|\theta - \theta^*\|_H^2\right]$ by our losses $\mathbb{E}\left[f(\theta) - f(\theta^*)\right]$ and $\mathbb{E}\left[(f(\theta) - f(\theta^*))^2\right]$.

**Lemma 12.** *Whatever the probability distribution on $\theta$:*

$$\mathbb{E}\left[\|\theta - \theta^*\|_H^2\right] \leq \frac{4\sigma}{1-\tilde{\eta}}\mathbb{E}\left[f(\theta) - f(\theta^*)\right] + \frac{10}{(1-\tilde{\eta})^2}\mathbb{E}\left[(f(\theta) - f(\theta^*))^2\right].$$

*Hence the iterates $(\theta_n)_{n\geq 0}$ following the SGD recursion from eq. (2) with step sizes $\gamma_n = \gamma_0/\sqrt{n}$ are such that:*

$$\mathbb{E}\left[\|\theta_n - \theta^*\|_H^2\right] \leq \frac{4\sigma}{1-\tilde{\eta}}\frac{\ln(en)}{\sqrt{n}}\left[\frac{3\|\theta_0 - \theta^*\|^2}{\gamma_0} + 4R^2\gamma_0\ln(en)\right] + \frac{80}{(1-\tilde{\eta})^2}\frac{\ln^2 en}{n}\left[4\frac{\|\theta_0 - \theta^*\|^2}{\gamma_0} + 20\gamma_0 R^2\ln en\right]^2.$$

*Proof.* The first part of the proof directly follows from Lemma 11:

$$\mathbb{E}\left[\|\theta - \theta^*\|_H^2\right] = \mathbb{E}\left[\|\theta - \theta^*\|_H^2 \, \mathbb{1}\{\|\theta - \theta^*\|_H \leq \sigma\}\right] + \mathbb{E}\left[\|\theta - \theta^*\|_H^2 \, \mathbb{1}\{\|\theta - \theta^*\|_H \geq \sigma\}\right]$$

$$\leq \frac{4\sigma}{1 - \tilde{\eta}} \mathbb{E}\left[(f(\theta) - f(\theta^*))\mathbb{1}\{\|\theta - \theta^*\|_H \leq \sigma\}\right] + \frac{10}{(1 - \tilde{\eta})^2} \mathbb{E}\left[(f(\theta) - f(\theta^*))^2 \mathbb{1}\{\|\theta - \theta^*\|_H \geq \sigma\}\right]$$

$$\leq \frac{4\sigma}{1 - \tilde{\eta}} \mathbb{E}\left[(f(\theta) - f(\theta^*))\right] + \frac{10}{(1 - \tilde{\eta})^2} \mathbb{E}\left[(f(\theta) - f(\theta^*))^2\right].$$

For the second part of the lemma we use the results from Lemma 10 to get:

$$\mathbb{E}\left[\|\theta_n - \theta^*\|_H^2\right] \leq \frac{4\sigma}{1 - \tilde{\eta}} \frac{\ln(en)}{\sqrt{n}} \left[\frac{3\|\theta_0 - \theta^*\|^2}{\gamma_0} + 4R^2\gamma_0 \ln(en)\right] + \frac{10}{(1 - \tilde{\eta})^2} \frac{8\ln^2 en}{n} \left[4\frac{\|\theta_0 - \theta^*\|^2}{\gamma_0} + 20\gamma_0 R^2 \ln en\right]^2.$$

$\square$

# C  Proof of the convergence guarantee.

In this section we prove the main result given Section 4. This first lemma is crucial and is the analogue of the self-concordance property from [2].

**Lemma 13.** *For all $z \in \mathbb{R}$:*

$$|\alpha(z) - \alpha(0)| \leq 20 \left(\ln \frac{2}{1 - \eta}\right) \frac{z}{\sigma} \alpha(z).$$

*Proof.* We proceed similarly as for Lemma 11. Notice that:

$$\alpha(z) = \sqrt{\frac{2}{\pi}} \frac{1}{\sqrt{\sigma^2 + z^2}} \left[(1 - \eta) + \eta \cdot \mathbb{E}_b\left[\exp\left(-\frac{b^2}{2(\sigma^2 + z^2)}\right) \mid b \neq 0\right]\right].$$

For $b \in \mathbb{R}$, let:

$$\alpha_b(z) = \sqrt{\frac{2}{\pi}} \frac{1}{\sqrt{\sigma^2 + z^2}} \left[(1 - \eta) + \eta \cdot \exp\left(-\frac{b^2}{2(\sigma^2 + z^2)}\right)\right],$$

so that $\alpha(z) = \mathbb{E}_b\left[\alpha_b(z) \mid b \neq 0\right]$. We render dimensionless the analysis by letting for $\tilde{b} \in \mathbb{R}^*$ and $\tilde{z} \in \mathbb{R}$:

$$\tilde{\alpha}_{\tilde{b}}(\tilde{z}) = \sqrt{\frac{2}{\pi}} \frac{1}{\sqrt{1 + \tilde{z}^2}} \left[(1 - \eta) + \eta \cdot \exp\left(-\frac{\tilde{b}^2}{1 + \tilde{z}^2}\right)\right].$$

Notice that $\alpha_b(z) = \frac{1}{\sigma} \tilde{\alpha}_{\tilde{b}}(\tilde{z})$ where $\tilde{z} = z/\sigma$ and $\tilde{b} = b/\sqrt{2}\sigma$.

Let $g(\tilde{z}) = \frac{\tilde{\alpha}_{\tilde{b}}(0)}{\tilde{\alpha}_{\tilde{b}}(\tilde{z})}$, notice that if we upper bound $|g'(\tilde{z})|$ by $20\left(\ln \frac{2}{1-\eta}\right)$. Then by a Taylor expansion we get that $|g(\tilde{z}) - g(0)| \leq 20\left(\ln \frac{2}{1-\eta}\right)\tilde{z}$ which will lead to the desired result.

Quick computations lead to:

$$g'(\tilde{z}) = \frac{(1 - \eta) + \eta \exp\left(-\tilde{b}^2\right)}{(1 - \eta) + \eta \exp\left(-\frac{\tilde{b}^2}{1+\tilde{z}^2}\right)} \frac{\tilde{z}}{\sqrt{1 + \tilde{z}^2}} \left[1 - 2\eta \frac{\tilde{b}^2}{1 + \tilde{z}^2} \frac{1}{(1 - \eta) \exp\left(\frac{\tilde{b}^2}{1+\tilde{z}^2}\right) + \eta}\right].$$

Notice that: $0 \leq \frac{(1-\eta)+\eta\exp\left(-\tilde{b}^2\right)}{(1-\eta)+\eta\exp\left(-\frac{\tilde{b}^2}{1+\tilde{z}^2}\right)} \leq 1$ and $0 \leq \frac{\tilde{z}}{\sqrt{1+\tilde{z}^2}} \leq 1$. Furthermore, from Lemma 17, for all $u \geq 0$: $\frac{u}{(1-p)+(1-\eta)\exp(u)} \leq 9\ln\frac{2}{1-\eta}$. Hence:

$$|g'(\tilde{z})| \leq 1 + 18\eta \ln \frac{2}{1 - \eta}$$

$$\leq 20 \ln \frac{2}{1 - \eta}.$$

Therefore, for all positive $\tilde{z}$, $|g(\tilde{z}) - g(0)| \leq 20\left(\ln\frac{2}{1-\eta}\right)\tilde{z}$. This implies that for all positive $\tilde{z}$,
$|\tilde{\alpha}_{\tilde{b}}(\tilde{z}) - \tilde{\alpha}_{\tilde{b}}(0)| \leq 20\left(\ln\frac{2}{1-\eta}\right)\tilde{z}\tilde{\alpha}_{\tilde{b}}(\tilde{z})$.

Now for $z, b \geq 0$ let $\tilde{z} = z/\sigma$ and $\tilde{b} = b/\sigma$:

$$
\begin{aligned}
|\alpha_b(z) - \alpha_b(0)| &= \frac{1}{\sigma}\left|\tilde{\alpha}_{\tilde{b}}(\tilde{z}) - \tilde{\alpha}_{\tilde{b}}(0)\right| \\
&\leq \frac{20}{\sigma}\left(\ln\frac{2}{1-\eta}\right)\tilde{z}\tilde{\alpha}_{\tilde{b}}(\tilde{z}) \\
&= 20\left(\ln\frac{2}{1-\eta}\right)\frac{z}{\sigma}\alpha(z).
\end{aligned}
$$

Taking the expectation over $b \neq 0$ and using Jensen's inequality concludes the proof. □

The following lemma shows that $f$'s particular structural enables us to bound the distance between $\bar{\theta}_n$ for any sequence $(\theta_i)_{i=0}^{n-1}$ and the minimum $\theta^*$.

**Lemma 14.** *Let (A.1, A.2, A.3, A.4) hold. Then for any sequences $(\theta_i)_{i=0}^{n-1} \in \mathbb{R}^{dn}$ their average $\bar{\theta}_n = \frac{1}{n}\sum_{i=0}^{n-1}\theta_i$ satisfies:*

$$
\mathbb{E}\left[\|\bar{\theta}_n - \theta^*\|_H^2\right] \leq \frac{2\sigma^2}{(1-\tilde{\eta})^2}\mathbb{E}\left[\left\|\frac{1}{n}\sum_{i=0}^{n-1}f'(\theta_i)\right\|_{H^{-1}}^2\right] + \frac{800}{(1-\tilde{\eta})^2}\left(\ln\frac{2}{1-\eta}\right)^2\mathbb{E}\left[\left(\frac{1}{n}\sum_{k=0}^{n-1}\langle f'(\theta_i), \theta_i - \theta^*\rangle\right)^2\right].
$$

*Proof.* In the following inequalities, we first use that $\|\cdot\|_{H^{-1}}$ is a norm and then use Lemma 13 with $z = \|\theta_i - \theta^*\|_H$.

$$
\begin{aligned}
\left\|\frac{1}{n}\sum_{i=0}^{n-1}f'(\theta_i) - f''(\theta^*)(\bar{\theta}_n - \theta^*)\right\|_{H^{-1}} &= \left\|\frac{1}{n}\sum_{i=0}^{n-1}(f'(\theta_i) - f''(\theta^*)(\theta_i - \theta^*))\right\|_{H^{-1}} \\
&\leq \frac{1}{n}\sum_{i=0}^{n-1}\|f'(\theta_i) - f''(\theta^*)(\theta_i - \theta^*)\|_{H^{-1}} \\
&= \frac{1}{n}\sum_{i=0}^{n-1}|\alpha(\|\theta_i - \theta^*\|_H) - \alpha(0)|\,\|H(\theta_i - \theta^*)\|_{H^{-1}} \\
&\leq \frac{20}{\sigma}\left(\ln\frac{2}{1-\eta}\right)\frac{1}{n}\sum_{i=0}^{n-1}\alpha(\|\theta_i - \theta^*\|_H)\|\theta_i - \theta^*\|_H^2 \\
&= \frac{20}{\sigma}\left(\ln\frac{2}{1-\eta}\right)\frac{1}{n}\sum_{k=0}^{n-1}\langle f'(\theta_i), \theta_i - \theta^*\rangle.
\end{aligned}
$$

Hence:

$$
\|f''(\theta^*)(\bar{\theta}_n - \theta^*)\|_{H^{-1}}^2 \leq 2\left\|\frac{1}{n}\sum_{i=0}^{n-1}f'(\theta_i)\right\|_{H^{-1}}^2 + \frac{800}{\sigma^2}\left(\ln\frac{2}{1-\eta}\right)^2\left(\frac{1}{n}\sum_{k=0}^{n-1}\langle f'(\theta_i), \theta_i - \theta^*\rangle\right)^2.
$$

Since $f''(\theta^*) = \sqrt{\frac{2}{\pi}}\frac{1-\tilde{\eta}}{\sigma}H$, we get:

$$
\mathbb{E}\left[\|\bar{\theta}_n - \theta^*\|_H^2\right] \leq \frac{\sigma^2}{(1-\tilde{\eta})^2}\left(2\mathbb{E}\left[\left\|\frac{1}{n}\sum_{i=0}^{n-1}f'(\theta_i)\right\|_{H^{-1}}^2\right] + \frac{800}{\sigma^2}\left(\ln\frac{2}{1-\eta}\right)^2\mathbb{E}\left[\left(\frac{1}{n}\sum_{k=0}^{n-1}\langle f'(\theta_i), \theta_i - \theta^*\rangle\right)^2\right]\right),
$$

which ends the proof of the lemma. □

We now show that $\left\|\bar{f}'(\theta_n)\right\|^2$, the square norm of the average of the gradients, converges at rate $O(1/n)$.

**Lemma 15.** *Let (A.1, A.2, A.3, A.4) and consider the SGD iterates following Eq. (2). Assume* $\gamma_n = \frac{\gamma_0}{\sqrt{n}}$. *Then for all $n \geq 1$ :*

$$\mathbb{E}\left[\left\|\frac{1}{n}\sum_{i=0}^{n-1}f'(\theta_i)\right\|_{H^{-1}}^2\right] \leq \frac{16d}{n} + \frac{4}{n\gamma_0^2}\mathbb{E}\left[\|\theta_n - \theta^*\|_{H^{-1}}^2\right] + \frac{4}{n^2\gamma_0^2}\|\theta_0 - \theta^*\|_{H^{-1}}^2$$

$$+ \frac{4}{n^2}\left(\sum_{i=1}^{n-1}\mathbb{E}\left[\|\theta_i - \theta^*\|_{H^{-1}}^2\right]^{1/2}\left(\frac{1}{\gamma_{i+1}} - \frac{1}{\gamma_i}\right)\right)^2.$$

*Proof.* Starting from the SGD recursion for $i \geq 1$:

$$\theta_i = \theta_{i-1} - \gamma_i f_i'(\theta_{i-1})$$
$$= \theta_{i-1} - \gamma_i f'(\theta_i) + \gamma_i \varepsilon_i(\theta_{i-1}),$$

where $\varepsilon_i(\theta_{i-1}) = f'(\theta_{i-1}) - \text{sgn}(\langle x_i, \theta_{i-1}\rangle - y_i)x_i$. Hence by rearranging we get that $f'(\theta_{i-1}) = \frac{\delta_{i-1}-\delta_i}{\gamma_i} + \varepsilon_i(\theta_{i-1})$. We sum from $1$ to $n$ to obtain:

$$\sum_{i=0}^{n-1}f'(\theta_i) = \frac{\|\theta_0 - \theta^*\|}{\gamma_0} - \frac{\|\theta_n - \theta^*\|}{\gamma_n} + \sum_{i=1}^{n-1}\|\theta_i - \theta^*\|\left(\frac{1}{\gamma_{i+1}} - \frac{1}{\gamma_i}\right) + \sum_{i=0}^{n-1}\varepsilon_{i+1}(\theta_i).$$

Note that $\|\,.\,\|_{H^{-1}}$ is a norm, hence:

$$\left\|\sum_{i=0}^{n-1}f'(\theta_i)\right\|_{H^{-1}} \leq \frac{1}{\gamma_0}\|\theta_0 - \theta^*\|_{H^{-1}} + \frac{1}{\gamma_n}\|\theta_n - \theta^*\|_{H^{-1}} + \sum_{i=1}^{n-1}\|\theta_i - \theta^*\|_{H^{-1}}\left(\frac{1}{\gamma_{i+1}} - \frac{1}{\gamma_i}\right) + \left\|\sum_{i=0}^{n-1}\varepsilon_{i+1}(\theta_i)\right\|_{H^{-1}}.$$

Using Minkowski's inequality we obtain:

$$\mathbb{E}\left[\left\|\sum_{i=0}^{n-1}f'(\theta_i)\right\|_{H^{-1}}^2\right] \leq \frac{4}{\gamma_0^2}\mathbb{E}\left[\|\theta_0 - \theta^*\|_{H^{-1}}^2\right] + \frac{4}{\gamma_n^2}\mathbb{E}\left[\|\theta_n - \theta^*\|_{H^{-1}}^2\right]$$

$$+ 4\left(\sum_{i=1}^{n-1}\mathbb{E}\left[\|\theta_i - \theta^*\|_{H^{-1}}^2\right]^{1/2}\left(\frac{1}{\gamma_{i+1}} - \frac{1}{\gamma_i}\right)\right)^2 + 4\mathbb{E}\left[\left\|\sum_{i=0}^{n-1}\varepsilon_{i+1}(\theta_i)\right\|_{H^{-1}}^2\right].$$

We now bound the sum of noises. Since $\mathbb{E}\left[\varepsilon_{i+1}(\theta_i)\,|\,\mathcal{F}_i\right] = 0$, using classical martingale second moment expansions:

$$\mathbb{E}\left[\left\|\sum_{i=0}^{n-1}\varepsilon_{i+1}(\theta_i)\right\|_{H^{-1}}^2\right] = \sum_{i=0}^{n-1}\mathbb{E}\left[\|\varepsilon_{i+1}(\theta_i)\|_{H^{-1}}^2\right]$$

$$\leq 2\sum_{i=0}^{n-1}\left(\mathbb{E}\left[\|f'(\theta_i)\|_{H^{-1}}^2\right] + \mathbb{E}\left[\|x\|_{H^{-1}}^2\right]\right).$$

Notice that $\mathbb{E}\left[\|x\|_{H^{-1}}^2\right] = d$. Furthermore, since $f'(\theta) = \alpha(\|\theta - \theta^*\|_H)\,H(\theta - \theta^*)$, we obtain $\|f'(\theta)\|_{H^{-1}}^2 = \alpha(\|\theta - \theta^*\|_H)^2\|\theta - \theta^*\|_H^2 \leq 2/\pi \leq 1$. Hence:

$$\mathbb{E}\left[\left\|\sum_{i=0}^{n-1}\varepsilon_{i+1}(\theta_i)\right\|_{H^{-1}}^2\right] \leq 2n(d+1) \leq 4nd.$$

This proves the lemma. $\qquad\square$

**Proof of Theorem 4.**

**Theorem 16.** *Let (A.1, A.2, A.3, A.4) hold and consider the SGD iterates following Eq. (2). Assume* $\gamma_n = \frac{\gamma_0}{\sqrt{n}}$. *Then for all* $n \geq 1$:

$$\mathbb{E}\left[\left\|\bar{\theta}_n - \theta^*\right\|_H^2\right] = O\left(\frac{\sigma^2 d}{(1-\tilde{\eta})^2 n}\right) + \tilde{O}\left(\frac{\|\theta_0 - \theta^*\|^4}{\gamma_0^2(1-\tilde{\eta})^2 n}\right) + \tilde{O}\left(\frac{\gamma_0^2 R^4}{(1-\tilde{\eta})^2 n}\right) + \tilde{O}\left(\frac{1}{\mu^2(1-\tilde{\eta})^3 n^{3/2}}\right).$$

*Proof.* To prove the final result it remains to upper bound $\frac{1}{\gamma_n}\mathbb{E}\left[\|\theta_n - \theta^*\|_{H^{-1}}^2\right]$ and $\sum_{i=1}^{n-1} \mathbb{E}\left[\|\theta_i - \theta^*\|_{H^{-1}}^2\right]^{1/2}\left(\frac{1}{\gamma_{i+1}} - \frac{1}{\gamma_i}\right)$ in Lemma 6.

To do so we upper bound $\mathbb{E}\left[\|\theta_i - \theta^*\|_{H^{-1}}^2\right]$ by $\frac{1}{\mu^2}\mathbb{E}\left[\|\theta_i - \theta^*\|_H^2\right]$ which can be upper bounded using Lemma 12:

$$\mathbb{E}\left[\|\theta_i - \theta^*\|_H^2\right] \leq \frac{4\sigma}{1-\tilde{\eta}}\frac{\ln(ei)}{\sqrt{i}}\left[\frac{3\|\theta_0 - \theta^*\|^2}{\gamma_0} + 4R^2\gamma_0\ln(ei)\right] + \frac{80}{(1-\tilde{\eta})^2}\frac{\ln^2 en}{n}\left[4\frac{\|\theta_0 - \theta^*\|^2}{\gamma_0} + 20\gamma_0 R^2\ln en\right]^2$$

$$\leq A_n\frac{1}{\sqrt{i}} + B_n\frac{1}{i},$$

where $A_n = \frac{4\sigma}{1-\tilde{\eta}}\ln(en)\left[\frac{3\|\theta_0-\theta^*\|^2}{\gamma_0} + 4\gamma_0 R^2\ln(en)\right]$, and $B_n = \frac{80}{(1-\tilde{\eta})^2}\ln^2(en)\left[4\frac{\|\theta_0-\theta^*\|^2}{\gamma_0} + 20\gamma_0 R^2\ln en\right]^2$. Therefore:

$$\sum_{i=1}^{n-1} \mathbb{E}\left[\|\theta_i - \theta^*\|_{H^{-1}}^2\right]^{1/2}\left(\frac{1}{\gamma_{i+1}} - \frac{1}{\gamma_i}\right) \leq \frac{1}{2\mu\gamma_0}\sum_{i=1}^{n-1}\left(\sqrt{A_n}\frac{1}{i^{1/4}} + \sqrt{B_n}\frac{1}{\sqrt{i}}\right)\frac{1}{\sqrt{i}}$$

$$\leq \frac{1}{2\mu\gamma_0}\left(4\sqrt{A_n}n^{1/4} + \sqrt{B_n}\ln en\right),$$

and:

$$\frac{1}{\gamma_n^2}\mathbb{E}\left[\|\delta_n\|_{H^{-1}}^2\right] \leq \frac{1}{\mu^2\gamma_0^2}\left(A_n\sqrt{n} + B_n\right).$$

We can then re-inject these bounds into Lemma 6:

$$\mathbb{E}\left[\left\|\frac{1}{n}\sum_{i=0}^{n-1} f'(\theta_i)\right\|_{H^{-1}}^2\right] \leq \frac{16d}{n} + \frac{4}{\mu n^2\gamma_0^2}\|\theta_0 - \theta^*\|^2 + \frac{4}{\mu^2 n^2\gamma_0^2}\left(A_n\sqrt{n} + B_n\right) + \frac{4}{\mu^2 n^2\gamma_0^2}\left(4A_n\sqrt{n} + B_n\ln^2 en\right)$$

$$\leq \frac{16d}{n} + \frac{4}{\mu n^2\gamma_0^2}\|\theta_0 - \theta^*\|^2 + \frac{4}{\mu^2 n^2\gamma_0^2}\left(5A_n\sqrt{n} + 2B_n\ln^2 en\right)$$

$$\leq \frac{16d}{n} + \frac{4}{\mu n^2\gamma_0^2}\|\theta_0 - \theta^*\|^2$$

$$+ \frac{4}{\mu^2 n^2\gamma_0^2}\left[5\frac{4\sigma}{1-\tilde{\eta}}\ln(en)\left[\frac{3\|\theta_0 - \theta^*\|^2}{\gamma_0} + 4\gamma_0 R^2\ln(en)\right]\sqrt{n}\right.$$

$$\left. + 2\frac{10}{(1-\tilde{\eta})^2}\frac{8\ln^2 en}{n}\left[4\frac{\|\theta_0 - \theta^*\|^2}{\gamma_0} + 20\gamma_0 R^2\ln en\right]^2\ln^2 en\right]$$

$$\leq \frac{16d}{n} + \frac{4}{\mu n^2\gamma_0^2}\|\theta_0 - \theta^*\|^2 + \frac{a_1(n)}{\mu^2(1-\tilde{\eta})n^{3/2}} + \frac{a_2(n)}{\mu^2(1-\tilde{\eta})^2 n^2},$$

where $a_1(n) = \frac{80\sigma\ln en}{\gamma_0^2}\left[\frac{3\|\theta_0-\theta^*\|^2}{\gamma_0} + 4\gamma_0 R^2\ln(en)\right]$ and $a_2(n) = \frac{640\ln^4 en}{\gamma_0^2}\left[\frac{4\|\theta_0-\theta^*\|^2}{\gamma_0} + 20\gamma_0 R^2\ln(en)\right]^2$.

We can now inject this bound along with Lemma 9 in Lemma 14:

$$\mathbb{E}\left[\|\bar{\theta}_n - \theta^*\|_H^2\right] \leq \frac{2\sigma^2}{(1-\tilde{\eta})^2}\mathbb{E}\left[\left\|\frac{1}{n}\sum_{i=0}^{n-1}f'(\theta_i)\right\|_{H^{-1}}^2\right] + \frac{800}{(1-\tilde{\eta})^2}\left(\ln\frac{2}{1-\eta}\right)^2\mathbb{E}\left[\left(\frac{1}{n}\sum_{k=0}^{n-1}\langle f'(\theta_i),\,\theta_i-\theta^*\rangle\right)^2\right]$$

$$\leq \frac{32\sigma^2 d}{(1-\tilde{\eta})^2 n} + \frac{8\sigma^2}{\mu(1-\tilde{\eta})^2 n^2\gamma_0^2}\|\theta_0-\theta^*\|^2 + \frac{2\sigma a_1(n)}{\mu^2(1-\tilde{\eta})^3 n^{3/2}} + \frac{2\sigma a_2(n)}{\mu^2(1-\tilde{\eta})^4 n^2}$$

$$+ \frac{800}{(1-\tilde{\eta})^2}\left(\ln\frac{2}{1-\eta}\right)^2\frac{\ln(en)}{n}\left[\frac{\|\theta_0-\theta^*\|^2}{\gamma_0} + 6\gamma_0 R^2\ln(en)\right]^2$$

$$\leq \frac{32\sigma^2 d}{(1-\tilde{\eta})^2 n} + \frac{1600}{(1-\tilde{\eta})^2}\left(\ln\frac{2}{1-\eta}\right)^2\frac{\ln en}{n}\left[\frac{\|\theta_0-\theta^*\|^4}{\gamma_0^2} + 36\gamma_0^2 R^4\ln^2(en)\right]$$

$$+ \frac{2\sigma a_1(n)}{\mu^2(1-\tilde{\eta})^3 n^{3/2}} + \frac{2\sigma a_2(n)}{\mu^2(1-\tilde{\eta})^4 n^2} + \frac{8\sigma^2}{\mu(1-\tilde{\eta})^2 n^2\gamma_0^2}\|\theta_0-\theta^*\|^2$$

$$= O\left(\frac{\sigma^2 d}{(1-\tilde{\eta})^2 n}\right) + \tilde{O}\left(\frac{\|\theta_0-\theta^*\|^4}{\gamma_0^2(1-\tilde{\eta})^2 n}\right) + \tilde{O}\left(\frac{\gamma_0^2 R^4}{(1-\tilde{\eta})^2 n}\right)$$

$$+ \tilde{O}\left(\frac{\sigma^2}{\gamma_0^2\mu^2(1-\tilde{\eta})^3 n^{3/2}}\left(\frac{\|\theta_0-\theta^*\|^2}{\gamma_0} + \gamma_0 R^2\right)\right).$$

$\square$

**Remark.** Notice that we could have also bounded $\mathbb{E}\left[\|\theta_{i+1}-\theta^*\|_{H^{-1}}^2\right]$ differently, since from eq. (2):

$$\mathbb{E}\left[\|\theta_{i+1}-\theta^*\|_{H^{-1}}^2\right] = \mathbb{E}\left[\|\theta_i-\theta^*\|_{H^{-1}}^2\right] + \gamma_{i+1}^2 d - \gamma_{i+1}\langle\theta_i-\theta^*,\,f'(\theta_i)\rangle_{H^{-1}}$$

Notice that $\langle\theta_i-\theta^*,\,f'(\theta_i)\rangle_{H^{-1}} = \alpha(\|\theta_i-\theta^*\|_H)\|\theta_i-\theta^*\|^2 \geq 0$. Thus $\mathbb{E}\left[\|\theta_{i+1}-\theta^*\|_{H^{-1}}^2\right] \leq \mathbb{E}\left[\|\theta_i-\theta^*\|_{H^{-1}}^2\right] + \gamma_{i+1}^2 d$ and $\mathbb{E}\left[\|\theta_i-\theta^*\|_{H^{-1}}^2\right]^{1/2} \leq \mathbb{E}\left[\|\theta_0-\theta^*\|_{H^{-1}}^2\right]^{1/2} + \gamma_0\sqrt{d}\sqrt{\ln(ei)}$.
Hence:

$$\sum_{i=1}^{n-1}\mathbb{E}\left[\|\theta_i-\theta^*\|_{H^{-1}}^2\right]^{1/2}\left(\frac{1}{\gamma_{i+1}}-\frac{1}{\gamma_i}\right) \leq \mathbb{E}\left[\|\theta_0-\theta^*\|_{H^{-1}}^2\right]^{1/2}\left(\frac{1}{\gamma_n}-\frac{1}{\gamma_0}\right)$$

$$+ \sqrt{d}\sum_{i=1}^{n-1}\sqrt{\ln(ei)}(\sqrt{i+1}-\sqrt{i})$$

$$\leq \frac{\mathbb{E}\left[\|\theta_0-\theta^*\|_{H^{-1}}^2\right]^{1/2}}{\gamma_n} + \sqrt{d\ln(en)n}.$$

This leads to a simpler upperbound on $\mathbb{E}\left[\left\|\frac{1}{n}\sum_{i=0}^{n-1}f'(\theta_i)\right\|_{H^{-1}}^2\right]$:

$$\mathbb{E}\left[\left\|\frac{1}{n}\sum_{i=0}^{n-1}f'(\theta_i)\right\|_{H^{-1}}^2\right] \leq \left(\frac{16}{n\gamma_0^2}\|\theta_0-\theta^*\|_{H^{-1}}^2 + \frac{32\ln(en)d}{n}\right)$$

The bias term is here $O(1/\mu n)$ instead of $O(1/\mu^2 n^{3/2})$.

## D  Technical lemmas.

In this section we prove a few technical lemmas. The first lemma is useful for the proof of Lemma 13.

**Lemma 17.** *For all $\eta \in [0,1)$ and $u \geq 0$:*

$$\frac{u}{\eta + (1-\eta)\exp(u)} \leq 9\ln\frac{2}{1-\eta}.$$

*Proof.* For $(1-\eta) \in (0,1)$ let $h(u) = \frac{u}{\eta+(1-\eta)\exp(u)}$. $h$ has a unique maximum on $\mathbb{R}_+$ which is attained in $u_c$ such that $h'(u_c) = 0$. This is equivalent to $\eta + (1-\eta)\exp(u_c) = (1-\eta)\exp(u_c)u_c$ and $(1-\eta)\exp(u_c) = \frac{\eta}{u_c-1}$. Hence for all $u \geq 0$, $h(u) \leq h(u_c) = \frac{1}{(1-\eta)\exp(u_c)} = \frac{u_c-1}{\eta}$. Furthermore, $(u_c-1)\exp(u_c-1) = \frac{1}{e}(\frac{1}{1-\eta}-1)$. Therefore $u_c - 1 = W\left(\frac{1}{e}\left(\frac{1}{1-\eta}-1\right)\right)$ where $W$ is the Lambert function and $h(u_c) = \frac{W\left(\frac{1}{e}(\frac{1}{1-\eta}-1)\right)}{\eta}$. Classical results on the Lambert function give that for $x \geq 1$, $W(x) \leq \ln x$ and for $x \geq 0$, $W(x) \leq x$.

Hence for $(1-\eta) \in (0, \frac{1}{1+e^2}]$, then $\frac{1}{e}(\frac{1}{1-\eta}-1) \geq 1$ therefore $W\left(\frac{1}{e}(\frac{1}{1-\eta}-1)\right) \leq \ln\frac{1}{e}(\frac{1}{1-\eta}-1) \leq \ln\frac{2}{1-\eta}$ and $h(u_c) \leq \frac{1}{\eta}\ln\frac{2}{1-\eta} = \left(\frac{1}{e^2}+1\right)\ln\frac{2}{1-\eta} \leq 9\ln\frac{2}{1-\eta}$.

For $(1-\eta) \in [\frac{1}{1+e^2}, 1)$, $W\left(\frac{1}{e}(\frac{1}{1-\eta}-1)\right) \leq \frac{1}{e}(\frac{1}{1-\eta}-1)$ and $h(u_c) \leq \frac{1}{(1-\eta)e} \leq \frac{1+e^2}{e} \leq 9\ln 2 \leq 9\ln\frac{2}{1-\eta}$.

For $\eta = 0$, $u_c = 1$, $h(u_c) = e^{-1}$ and the inequality still holds. $\square$

The two following lemmas are used in Lemma 11.

**Lemma 18.** *For all $x \geq 0$ :*

$$\frac{\exp(-x^2)}{5} \leq \frac{\sqrt{2}-1}{\sqrt{\pi}}\exp(-x^2) \leq x\left(\mathrm{erf}\left(\frac{x}{\sqrt{2}}\right) - \mathrm{erf}(x)\right) + \sqrt{\frac{2}{\pi}}\exp\left(-\frac{x^2}{2}\right) - \frac{1}{\sqrt{\pi}}\exp(-x^2).$$

*Proof.* Let $h(x) = x\left(\mathrm{erf}\left(\frac{x}{\sqrt{2}}\right) - \mathrm{erf}(x)\right) + \sqrt{2/\pi}\left(\exp\left(-\frac{x^2}{2}\right) - \exp(-x^2)\right)$. We show that $h(x) \geq 0$ which proves the lemma. We compute the first and second derivative of $h$, which leads to

$$h'(x) = \mathrm{erf}\left(\frac{x}{\sqrt{2}}\right) - \mathrm{erf}(x) + x\exp(-x^2)\frac{2}{\sqrt{\pi}}(\sqrt{2}-1),$$

$$h''(x) = \sqrt{\frac{2}{\pi}}\exp(-x^2)\left[\exp\left(\frac{x^2}{2}\right) - \sqrt{2} + \sqrt{2}(\sqrt{2}-1)(1-2x^2)\right].$$

Notice that the zeros of $h''$ on $\mathbb{R}_+$ correspond to the intersection of an exponential and an upward parabola: there are only 2 which we call $0 < x_1 < x_2$. Also note that $h''$ is strictly positive on $(0, x_1) \cup (x_2, +\infty)$ and strictly negative on $(x_1, x_2)$. Since $h'(0) = 0$ and $h' \underset{x\to+\infty}{\to} 0$ we have that $h'$ has only one zero on $R_+^*$ which we denote $x_c$ and: $h'$ is positive on $[0, x_c]$, negative on $[x_c, +\infty)$. Since $h(0) = 0$ and $h \underset{x\to+\infty}{\to} 0$ we conclude that $h$ is positive on $\mathbb{R}_+$.

$\square$

**Lemma 19.** *Let $\tilde{f}_{\tilde{b}}(\tilde{\sigma}) = \tilde{b}\,\mathrm{erf}\left(\frac{\tilde{b}}{\sqrt{1+\tilde{\sigma}^2}}\right) + \frac{1}{\sqrt{\pi}}\sqrt{1+\tilde{\sigma}^2}\exp\left(-\frac{\tilde{b}^2}{1+\tilde{\sigma}^2}\right)$. For all $\tilde{b} \in \mathbb{R}$ and $\tilde{\sigma} \leq 1$,*

$$\frac{\tilde{\sigma}^2}{5}\exp\left(-\tilde{b}^2\right) \leq \frac{\sqrt{2}-1}{\sqrt{\pi}}\tilde{\sigma}^2\exp\left(-\tilde{b}^2\right) \leq \tilde{f}_{\tilde{b}}(\tilde{\sigma}) - \tilde{f}_{\tilde{b}}(0).$$

*Proof.* Let $\tilde{b} \in \mathbb{R}$ and consider for $x \in [0,1]$, $h(x) = \tilde{b} \operatorname{erf}\left(\frac{\tilde{b}}{\sqrt{1+x}}\right) + \frac{1}{\sqrt{\pi}}\sqrt{1+x}\exp\left(-\frac{\tilde{b}^2}{1+x}\right) - \frac{\sqrt{2}-1}{\sqrt{\pi}}x\exp\left(-\tilde{b}^2\right)$. Then: $h'(x) = \frac{1}{2\sqrt{\pi}\sqrt{1+x}}\exp\left(-\frac{\tilde{b}^2}{1+x}\right) - \frac{\sqrt{2}-1}{\sqrt{\pi}}\exp\left(-\tilde{b}^2\right)$. We have $h''(x) = \frac{\exp\left(-\frac{\tilde{b}^2}{1+x}\right)}{4\sqrt{\pi}(1+x)^{5/2}}(2\tilde{b}^2 - (1+x))$.

- Therefore if $|\tilde{b}| \geq 1$ then $h''(x) \geq 0$ on $[0,1]$ and $h'$ is increasing on $[0,1]$. Therefore $h'(x) \geq h'(0) = \left(\frac{1}{2\sqrt{\pi}} - \frac{\sqrt{2}-1}{\sqrt{\pi}}\right)\exp\left(-\tilde{b}^2\right) > 0$. Hence $h$ is increasing and $h(x) \geq h(0)$.

- If $|\tilde{b}| \in [1/\sqrt{2}, 1]$, then for $x_0 = 2\tilde{b}^2 - 1$, $h''(x_0) = 0$, $h'$ is increasing then decreasing and $h'(x) \geq \min\{h'(0), h'(1)\} \geq 0$. Note that for $|\tilde{b}| \geq 1/\sqrt{2}$, $h'(1) \geq 0$. Hence for all $x \in [0,1]$, $h'(x) \geq 0$, therefore $h$ is increasing and $h(x) \geq h(0)$.

- Finally if $|\tilde{b}| \leq 1/\sqrt{2}$ then $h''(x) \leq 0$. Therefore $h$ is concave on $[0,1]$ and $h(x) \geq \min\{h(0), h(1)\}$. However notice that by Lemma 18 we have that $h(0) \leq h(1)$. Therefore $h(x) \geq h(0)$ on $[0,1]$.

Hence in all cases $h(x) \geq h(0)$ on $[0,1]$. Considering $x = \tilde{\sigma}^2$ concludes the proof. $\qquad\square$

This final lemma is required Lemma 10.

**Lemma 20.**
$$\frac{1}{n}\sum_{t=2}^{n-1}\frac{1}{\left(\frac{t}{n}\right)^2}\left((1-\frac{t}{n})^{-1/2} - 1\right) \leq 3\ln en.$$

*Proof.* For $0 < x < 1$ let $h(x) = \frac{1}{x^2}((1-x)^{-1/2} - 1)$.

We first show that $h(x) \leq \frac{1}{x} + (1-x)^{-1/2}$. Indeed, first notice that on $\mathbb{R}_+$, $h(x) = \frac{1}{x}$ has only one solution $x_c$ which is such that $1 = (x_c + 1)\sqrt{1-x_c}$, furthermore: $h(x) \underset{x\to 0}{\sim} \frac{1}{2x} \leq \frac{1}{x}$. Therefore by continuity hypotheses arguments, $h(x) \leq \frac{1}{x}$ on $(0, x_c]$. Similarly, $h(x) = \frac{1}{\sqrt{1-x}}$ has only one solution on $[0,1)$ which is also $x_c$ and $h(x) \underset{x\to 1}{=} \frac{1}{\sqrt{1-x}} - 1 + o(1)$, hence for $x$ close enough to 1 we have $h(x) \leq \frac{1}{\sqrt{1-x}}$ and by continuity arguments $h(x) \leq \frac{1}{\sqrt{1-x}}$ on $[x_c, 1)$. Finally we get that $h(x) \leq \frac{1}{x} + (1-x)^{-1/2}$ on $(0,1)$. We now use this bound to obtain the result:

$$\frac{1}{n}\sum_{t=2}^{n-1}\frac{1}{\left(\frac{t}{n}\right)^2}\left((1-\frac{t}{n})^{-1/2} - 1\right) \leq \frac{1}{n}\sum_{t=2}^{n-1}\left(\frac{1}{\left(\frac{t}{n}\right)} + (1-\frac{t}{n})^{-1/2}\right)$$
$$\leq \ln(en) + \int_0^1 (1-x)^{-1/2}dx$$
$$= \ln(en) + 2$$
$$\leq 3\ln(en).$$

$\qquad\square$

# E  Experiment Setup for the Breakdown Point Experiment

We followed the experimental setup of [79]. For Torrent, CRR and AdaCRR we used the implementations provided by the authors. We additionally used the matlab in-built implementation for the Huber regression. The hyperparameters of these algorithm were set by grid-search, except for AdaCRR for which they were set to their default values provided by [79]. To ensure that saturation was reached, we did 10 passes on the whole data when using our algorithm on the $\ell_1$-loss.