[Reviews · NeurIPS 2020]

Review 1

Summary and Contributions: This paper consider the problem of robust regression, and consider the simple algorithm: run SGD on ell_1 loss. And the authors establish a convergence rate on this simple algorithm: 1((1-eta)^2*n), though the dependence on eta might not be optimal, given the simplicity of the algorithm and strong empirical performance, I think this result deserves presenting at the conference.

Strengths: Clear theoretical justification of the proposed approach with guarantees.

Weaknesses: No real data experiments to verify the effectiveness of the approach in real world.

Correctness: Theoretical results are clearly stated and seem correct. Empirical results look reasonable.

Clarity: This paper is well written.

Relation to Prior Work: Related work are extensively discussed.

Reproducibility: Yes

Additional Feedback:


Review 2

Summary and Contributions: This paper performed a theoretical analysis on the empirical loss minimization with absolute loss. In a setting they called the online oblivious response corruption, they proved the average of a sequence obtained by the SGD achieves convergence rate of 1/n.

Strengths: They show the proposed algorithm (averaging of a sequence of SGD) is highly scalable and optimal. They show other benefits of the proposed algorithm such as low dependency on noise level and feature confounding.

Weaknesses: Their analysis seems to strongly depend on the assumption that data is drawn from 0 mean gaussian distribution. When it is said 1/n is the optimal rate, one usually assumes much less.

Correctness: Their proof seems to be correct.

Clarity: Yes

Relation to Prior Work: The authors relates this work to others from both aspects of robust statistics and stochastic optimization.

Reproducibility: No

Additional Feedback:


Review 3

Summary and Contributions: The robust linear regression problem is studied in the online setting. Under certain conditions, the convergence rate of the averaged iterate is obtained.

Strengths: The online version of SGD for the robust linear regression problem seems novel.

Weaknesses: The linear model is simple, and the assumptions on the data seems a little bit restrictive.

Correctness: The proofs seems correct, but the referee did not check them line by line.

Clarity: yes

Relation to Prior Work: seems so

Reproducibility: Yes

Additional Feedback: After the rebuttal: I am not very familiar with this area, while the authors addressed my concerns partially. Hence, I decide to remain the current score.


Review 4

Summary and Contributions: This paper addresses the task of online learning for robust linear regression (with L1 loss). In particular, based on the smoothing mechanism, the authors propose a stochastic gradient descent on the l1 loss with guaranteed convergence. The authors show some encouraging results on robust regression.

Strengths: - The problem of online learning for robust regression is quite important in several machine learning tasks, where the algorithm only has access to the the data in a streaming manner. Thus, the proposed algorithm would be very useful for such applications. - The non-smothness of the l1 loss is addressed using Gaussian smoothing. Although Gaussian smoothing is not new, using it in this online learning context is rather interesting. - The final algorithm is quite simple, which can be easily implemented using existing optimization frameworks. - A proof for convergence guarantee is provided.

Weaknesses: - As can be seen from Lemma 3, the higher the outlier proportion, the local conditioning becomes worse. Also, if the noise level tends to 0, the problem becomes non-smooth. It is not clear if the proposed algorithm still well-behaves in such extreme cases. - Some work on robust online regression had been previously discussed, for example: + Briegel, Thomas, and Volker Tresp. "Robust neural network regression for offline and online learning." Advances in Neural Information Processing Systems. 2000. It is not clear why such references are not mentioned and compared in the paper. ======== Update after rebuttal: I thank the authors for providing feedback to all reviewers' comments. Some of my concerns have been addressed. Therefore, I have upgraded my rating for this work to "7. Accept".

Correctness: - The proofs look reasonable (although the details have not been carefully checked). - The empirical results support the theory.

Clarity: - The paper is well written where most details are clearly explained

Relation to Prior Work: Some related works that addressed the similar problems are not thoroughly discussed.

Reproducibility: Yes

Additional Feedback: - The robustness of the algorithm is attained through L1 (or Huber loss). However, in many practical applications with high outlier rates, one needs to use more difficult loss functions, e.g., Tukey loss. It would be interesting to discuss if it is straight-forward to extend this work to such loss.

[Author Response · NeurIPS 2020]

We thank all the reviewers for their feedback and comments. They will be incorporated into the revised version.

We would like to highlight several points regarding our contribution. In the paper we consider the robust linear
regression problem which has drawn significant interest in the last decades. Even in the simpler setting where only the
responses are obliviously corrupted there have been numerous works proposing novel algorithms that recover the gold
model $\theta^*$. Though this setting may seem simple, it has considerable applications. Nonetheless, currently none of the
proposed algorithms are suitable for modern large-scale problems. This observation is the incentive behind our work:
we propose the first highly scalable and efficient algorithm for the robust linear regression problem with oblivious
response corruptions. We show that running SGD on the $\ell_1$ loss outperforms all current algorithms, theoretically
and empirically. Not only is it currently the unique suitable large-scale algorithm, but it also presents several major
advantages even in the offline setting: a) there are no hyperparameters to tune, the knowledge of $\sigma$ or $\eta$ are not necessary
b) the $O(1/n)$ dominant convergence terms are independent of the conditioning $\kappa$ of the data c) the algorithm adapts to
the difficulty of the adversary since the rate depends on the effective corruption proportion $\tilde{\eta}$. We also want to underline
that the algorithm is extremely simple to implement: it represents only two lines of code.

Concerning the Gaussian assumption on the data, we agree that it is restrictive and that it does not always represent real
world situations. However we highlight that this assumption (or similar specific structural assumptions on the design
matrix and the noise) is made in the previous papers that deal with the framework that we consider. The key for our
analysis is that the continuous density of $(x, \varepsilon)$ smooths the objective $f$, this enables a proof of type Polyak-Judistky to
obtain the final rate. The more restrictive Gaussian assumption is made in order to obtain a closed form for $f$ which
enables simpler (but not simple !) computations to obtain the final convergence rate. We believe that the proof could
be extended to any sub-Gaussian continuous data, however this would require considerably heavier and cumbersome
computations (see the figure for an example where the covariates are sampled from a zero-mean uniform distribution
with covariance $H$ and the noise from a uniform distribution with variance $\sigma^2$).

**To Reviewer 1.** As pointed out, the dependency on $\eta$ (or rather $\tilde{\eta}$) we obtain
might not be optimal and is still an interesting open question. It is however the
best dependency which has been obtained since the effective outlier proportion
$\tilde{\eta}$ is strictly smaller than $\eta$. We agree that real data experiments would have been
a plus, they will be added to the revised version.

Online robust regression on non-Gaussian data. Same adversarial setting as in Figure 1 in the paper.

**To Reviewer 2.** The centered data assumption is a classical assumption which
is often made in machine learning, note that it is for instance made in previous
papers that deal with the framework we consider. This assumption is reasonable
since it is of common practice to pre-process and center the features. We highlight
the fact that even for the Gaussian linear regression problem the minimax rate of
estimation is $O(1/n)$, in the sense that no estimator that uses the same information
can improve upon this rate. Therefore, in the setting we consider, the rate we
obtain cannot be improved.

**To Reviewer 3.** The linear model is one of the simplest model we could have considered and linear regression is
certainly amongst the oldest and most fundamental statistical methods. However: (a) it is still intensively used in
practice and studied in theory, (b) more complex models can be seen as an extension of it and (c) there are still many
interesting open questions. Indeed even for this simple model there were still no efficient algorithms which could deal
with large datasets, hence the motivation of our work. The extension of the algorithm to broader and more general
models is undoubtedly interesting, however it is outside the scope of our present work.

**To Reviewer 4.** As seen in Theorem 4, the terms depending on the variance $\sigma$ go to 0 as $\sigma \to 0$. Hence the extreme
case where there is no 'nice' noise is not pathological and the algorithm still performs well. On the other hand, the
performance of the algorithm is indeed inevitably dependent of $\tilde{\eta}$ (and not on $\eta$ as for other robust algorithms): we
cannot expect any algorithm to behave well as $\tilde{\eta}$ goes to 1 since this corresponds to having all the data corrupted. The
dependency $(1 - \tilde{\eta})^{-2}$ we obtain is to our knowledge the best which has been obtained.

We thank the reviewer for the interesting missing reference which will be added to the revised version. The literature on
robust regression is vast and we have tried to be as exhaustive as possible. The mentioned paper proposes an interesting
EM-algorithm, however they do not give any convergence guarantee. Furthermore every M-step of their algorithm is
computationally heavy since it requires solving a weighted LS problem on the whole dataset, making it inappropriate
for large-scale problems.

As pointed out, the Tukey loss is often used to deal with corruptions, however its non-convexity makes it much harder
to analyse and optimise. Furthermore, as for the Huber loss, the Tukey loss requires tuning an extra parameter and we
believe it will not lead to better experimental results for the linear model with corrupted response. However the Tukey
loss (as well as other redescending M-estimators) is well-known for being more robust to corruption in the features $x$.
We let further investigations as future work.

[Meta-Review · NeurIPS 2020]

The paper concerns robust linear regression in the online setting, where the data follows a Gaussian linear model with corruptions. It is shown that the stochastic gradient descent on the absolute loss converges to the true parameter at a rate of order O(1/n). The paper received a universally positive evaluation from the reviewers, who acknowledged the novelty of the results, the theoretical justification of the proposed approach and the scalability of the algorithm. The main issue raised in the reviews is about quite restrictive assumptions on the data distribution (Gaussian linear model, and the centered data assumption).